# Nanotopography reveals metabolites that maintain the immunomodulatory phenotype of mesenchymal stromal cells

Ewan A. Ross[1,2], Lesley-Anne Turner[1], Hannah Donnelly[1], Anwer Saeed[3], Monica P. Tsimbouri[1], Karl V. Burgess[4], Gavin Blackburn[4], Vineetha Jayawarna[5], Yinbo Xiao[1], Mariana A. G. Oliva[5], Jennifer Willis[2], Jaspreet Bansal[2], Paul Reynolds[3], Julia A. Wells[6], Joanne Mountford[7], Massimo Vassalli[5], Nikolaj Gadegaard[3], Richard O. C. Oreffo[6], Manuel Salmeron-Sanchez[5] & Matthew J. Dalby[1] ✉

Mesenchymal stromal cells (MSCs) are multipotent progenitor cells that are of considerable clinical potential in transplantation and anti-inflammatory therapies due to their capacity for tissue repair and immunomodulation. However, MSCs rapidly differentiate once in culture, making their large-scale expansion for use in immunomodulatory therapies challenging. Although the differentiation mechanisms of MSCs have been extensively investigated using materials, little is known about how materials can influence paracrine activities of MSCs. Here, we show that nanotopography can control the immunomodulatory capacity of MSCs through decreased intracellular tension and increasing oxidative glycolysis. We use nanotopography to identify bioactive metabolites that modulate intracellular tension, growth and immunomodulatory phenotype of MSCs in standard culture and during larger scale cell manufacture. Our findings demonstrate an effective route to support large-scale expansion of functional MSCs for therapeutic purposes.

Research has provided scientists with a variety of materials for investigating the physical properties that stimulate mesenchymal stromal cell (MSC) differentiation[1–4]. Material properties which have been exploited include stiffness[1], adhesivity[2,3], chemistry[5] and nanotopography[4]. The use of these tools has revealed that MSC differentiation along the adipogenic pathway is characterised by cells with low adhesion, low cytoskeletal tension, and by the suppression of the bone-related transcription factor, Runt-related transcription factor

2 (RUNX2)[2,3,6]. By contrast, MSC osteogenic differentiation is characterised by cells with high levels of adhesion and intracellular tension, mediated by Rho-A kinase (ROCK), as well as by RUNX2 activation[2,3,6]. Using materials to control MSC differentiation will be of great value for both biomaterials and tissue engineering approaches.

The potential therapeutic applications of MSCs are now extending beyond not just for tissue engineering, but towards use of these progenitors for cellular therapies to treat autoimmune or inflammatory

[1]Centre for the Cellular Microenvironment, School of Molecular Biosciences, College of Medical, Veterinary and Life Sciences, Mazumdar-Shaw Advanced Research Centre, University of Glasgow, Glasgow G11 6EW, UK. [2]School of Biosciences, College of Health and Life Sciences, Aston University, Birmingham B4 7ET, UK. [3]Division of Biomedical Engineering, James Watt School of Engineering, University of Glasgow, Glasgow G12 8QQ, UK. [4]Glasgow Polyomics, Wolfson Wohl Cancer Research Centre, Garscube Campus, Bearsden, Glasgow G61 1QH, UK. [5]Centre for the Cellular Microenvironment, Division of Biomedical Engineering, James Watt School of Engineering, Mazumdar-Shaw Advanced Research Centre, University of Glasgow, Glasgow G11 6EW, UK. [6]Bone and Joint Research Group, Centre for Human Development, Stem Cells and Regeneration, Institute of Developmental Sciences, University of Southampton, Southampton SO16 6YD, UK. [7]Scottish National Blood Transfusion Service, Advanced Therapeutics, Jack Copland Centre, 52 Research Avenue North, Heriot Watt Research Park, Edinburgh EH14 4BE, UK. ✉e-mail: matthew.dalby@glasgow.ac.uk

conditions. To support this growing clinical demand, it is critical that we develop methods to grow sufficient numbers of high quality, stable MSCs. This is important because ex vivo, out of their niche, MSCs rapidly and spontaneously differentiate, mainly because they are primed to differentiate by the unfamiliar stiffness and chemistry of surfaces within culture flasks, which were developed for more phenotypically stable cells[7–10].

The major focus for MSC use as a biological therapeutic is in suppression of immune responses. Via paracrine signalling, MSCs can modulate immune responses and reduce inflammation[11]. This capability is currently under evaluation in human trials for graft vs host disease following haematopoietic stem cell transplantation[12], and in the co-transplantation of MSCs with islet cells in the treatment of diabetes[13,14]. However, it is important to note that large-scale MSC production, which is required to generate sufficient quantities of clinical grade cells, is offset by a reduction in their immunomodulatory capability, which typically occurs following their long-term culture[15]. Furthermore, the currently advocated MSC expansion protocol relies on the use of multi-layer, large surface-area, cell-culture ware[16].

A small range of materials that can promote prolonged in vitro MSC self-renewal have been identified by utilising nanotopography[8], surface chemistry[17], elasticity[18] and micro-contact printed adhesive islands[10]. These studies demonstrate that when adhesion is reduced relative to adhesion levels in MSCs cultured on polystyrene, but not reduced to the level that would promote adipogenesis, stromal cell multipotency can be retained[10,18,19]. However, understanding of biological mechanism is nascent and ability to influence MSC immunomodulatory capability is unknown.

In this study, we use defined nanotopographies to investigate if materials, such as those that have been successfully used to study MSC multipotency and differentiation, can be used to study their suppressive, immunomodulatory properties as well. We also investigate if materials that stimulate enhanced MSC immunomodulatory capacity can be used to dissect the molecular mechanisms involved. We hypothesise that such materials could be used to highlight phenotype-specific targets, providing a potential means by which to identify metabolites that might serve as biological small molecules that could help to address the major challenges in the therapeutic manufacture of MSCs.

## Results

### Nanotopography can maintain MSC immunomodulatory capacity

Nanotopography has been previously demonstrated to promote MSC multipotency in culture without suppressing cell growth; specifically, nanopits of 120 nm diameter, 100 nm depth and 300 nm centre-to-centre spacing within a square arrangement (SQ, Fig. 1a)[8]. To investigate if this surface can also retain MSC immunomodulatory capacity, a key functional requirement in order to define a cell as a stromal cell progenitor[20], a T cell proliferation assay was used[21]. In this assay, human peripheral blood mononuclear cells (PBMCs) were labelled with the intracellular proliferation dye CFSE (carboxyfluorescein succinimidyl ester) and were stimulated with phytohemagglutinin (a mitogenic lectin) and interleukin-2 (IL-2) to drive the proliferation of T cells. CFSE-labelled PBMCs were then co-cultured with primary human MSCs for 5 days. The ability of MSCs to reduce T cell proliferation was measured through the detection of CFSE dilution by flow cytometry (upon cell proliferation, each daughter cell contains half the amount of intracellular CFSE compared to the mother cell; see Fig. S1). Stimulated T cells in the absence of MSCs were used as a positive control and T cells cultured in the absence of any stimulation as a negative control.

Stro-1[+] selected, bone marrow-derived, human skeletal MSCs were cultured for 14 days on the SQ nanotopography pattern, which had been injection moulded into polycarbonate. As positive controls, we used a flat topography and an osteogenic-differentiation-promoting nanotopography[4] (both also fabricated in polycarbonate). The differentiating nanotopography is similar to SQ but pits are offset by up to ± 50 nm in the x and y axes from the centre position[4], and is termed near square (NSQ, Fig. 1b).

CFSE-labelled T cells were added to MSCs and co-cultured for five days, followed by flow cytometry analysis. This showed that while MSCs cultured on flat and NSQ nanotopographies displayed poor immunomodulatory capabilities, as measured by increasing numbers of higher-division T cells, T cells cultured with MSCs on the SQ surface underwent fewer divisions (Fig. 1c). This is supported by the reduced proliferative index of T cells co-cultured with MSCs on SQ (Fig. 1d), relative to the other nanotopographies, indicating that the suppressive capacity of MSCs on SQ is significantly increased ($p = 0.0245$).

We further assessed the longer-term growth and potential of the SQ surface to maintain MSCs immunomodulatory capacity by culturing on topographies for longer periods. After four weeks of culture, growth was measured and there was no significant difference in cell numbers on the SQ and control surfaces (Fig. 1e) demonstrating that the SQ surface does not suppress growth. Looking at T cell suppression after 4 and 6 weeks of MSC culture, it was seen that SQ educated MSCs were more immunomodulatory compared to flat controls, demonstrating that their suppressive multipotent phenotype is maintained during this prolonged period in culture compared to flat control surfaces (Fig. 1f). It should be noted that the total surface energy of the topographies showed no differences between the different patterns (Fig. S2). Therefore, physiological differences observed are through modulation of cell behaviour via their contact with the different surfaces and not due to an intrinsic difference in the properties of the materials/surfaces being used. These findings indicate that a material alone has been shown to enhance the immunomodulatory capability of MSCs.

A further correlate of MSC immunomodulatory potential is that therapeutically useful MSCs undergo apoptosis in vitro in response to activated immune cells; i.e. they are more killable[22]. We assessed levels of caspase 3 and 6 in MSCs as they have important roles in executing apoptosis[23] and observed that caspase levels were increased in response to the SQ topography (Fig. S3a). MSCs cultured on SQ or flat control topographies were exposed to activated T cells and the induction of apoptosis and necrosis measured. We observed that while there was no evidence of increased necrosis, that there was a trend towards elevated apoptosis for MSCs cultured on the SQ nanotopography with T-cell co-culture (Fig. S3b, c). Together, this data suggests the cells cultured on SQ would be primed to undergo apoptosis in vivo in contact with activated T cells at a site of inflammation, and thus be therapeutically effective.

It is hypothesised that multipotency in MSCs is maintained through intermediate levels of intracellular tension, slightly lower than that of fibroblasts[19,24], with ROCK being a central mediator of intracellular tension in MSCs[2,3]. Again, using phalloidin staining of actin cytoskeleton, we observe that on the NSQ, osteogenic surface supports better spread cells with greater stress fibre organisation than for cells cultured on SQ (Fig. S4). To investigate the potential functional link between intracellular stiffness and immunomodulation, we first evaluated cell stiffness of MSCs cultured on the different topographies (Fig. 1g). MSCs cultured on the naive promoting SQ surface had significantly lower stiffness. This lowering of cell stiffness for cells on SQ was also apparent in hypoxic conditions at 1% oxygen tension (Fig. S5a). This suggests that changes in cell stiffness are not directly linked to environmental oxygen tension. However, hypoxia itself is known to induce immunomodulation[25] and thus resulted in a reduction in T cell

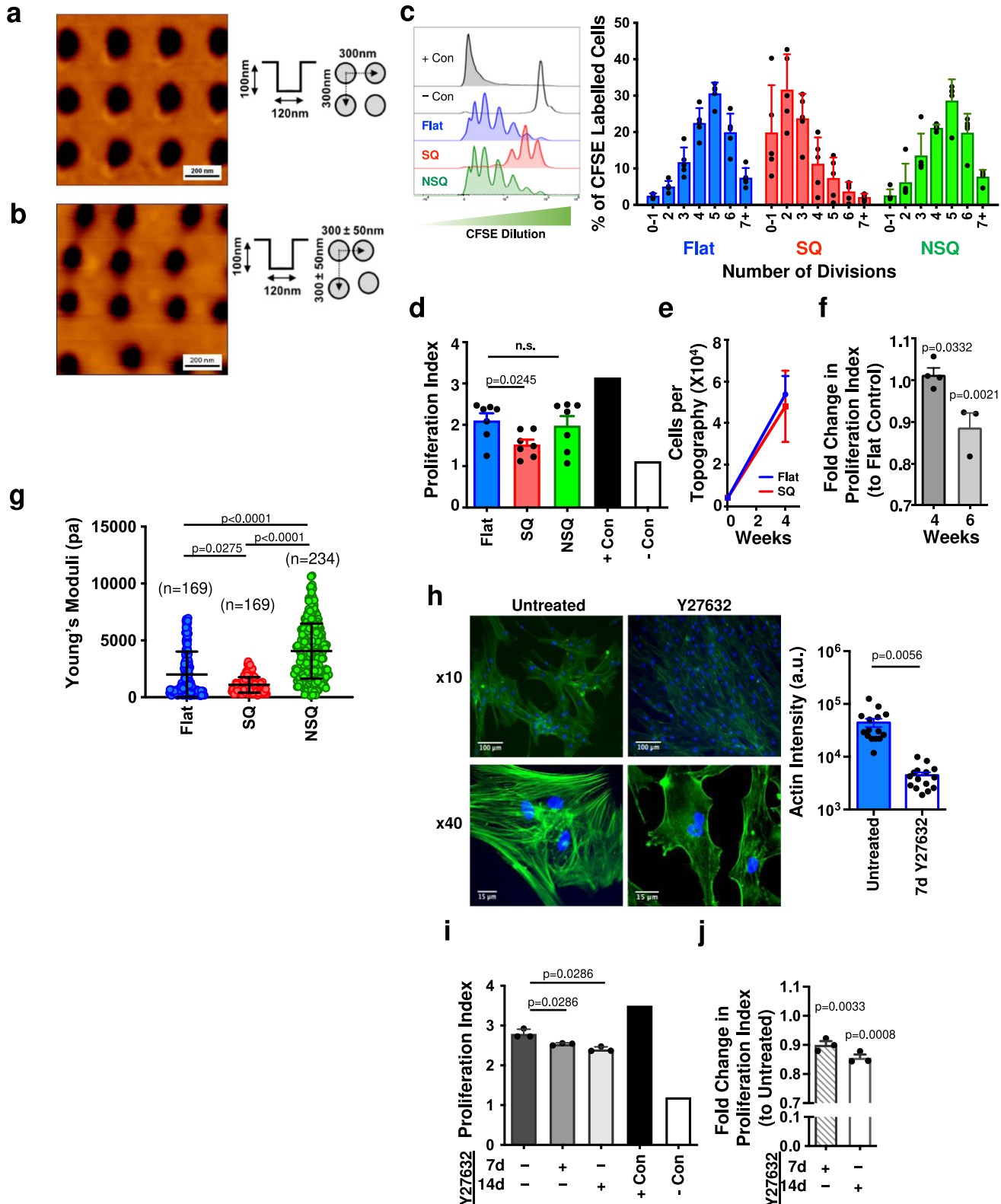

proliferation index to a level where the SQ surface contributed no further to immunomodulation (Fig. S5b, c).

Proposing that changes in intracellular tension are important to non-hypoxic regulation of immunomodulatory MSC phenotype, we hypothesised that reducing ROCK-mediated intracellular tension in MSCs on flat control surfaces, should drive these MSCs (which display a fibroblastic phenotypic drift, typical of MSCs in cell culture[8]), to behave more like immunomodulatory SQ cultured MSCs.

Morphological change was indeed observed (Fig. 1h) along with the reduction in the proliferation index of T cells co-cultured with ROCK inhibitor-treated MSCs on flat control surfaces (Fig. 1i, j). Thus, MSCs on flat controls with reduced intracellular tension display a more suppressive, immunomodulatory phenotype. These results support the hypothesis that the MSC phenotype is influenced by intracellular tension, with greater intracellular tension resulting in differentiation and reduced immunomodulatory capacity[24].

**Fig. 1 | Nanotopography can maintain MSC immunomodulatory capacity via decreased intracellular tension. a** Representative atomic force microscopy images of square (SQ) patterned and **b** osteogenic-enhancing, offset near square (NSQ) polycarbonate nanotopographies. **c** Stro-1⁺ MSCs were cultured on nanotopographies for 14 days, then co-cultured with CFSE-labelled, PHA and IL-2 stimulated PBMCs for a further 5 days. CFSE dilution was quantified by flow cytometry (left panel) and graph shows representative results from one co-culture (n = 4 topographies per group, mean ± S.D.). **d** Proliferation index was calculated to allow comparison of MSCs immunomodulatory potential from multiple donors (n = 7 donors). **e** Over longer, 4 week, culture, no change in cell growth was observed. **f** MSCs were cultured on SQ versus flat topographies for 4 or 6 weeks and effects on the proliferation index of co-cultured CFSE-labelled PBMCs assessed. In **e** and **f**, n = 4 topographies per group from one donor; representative of two independent experiments. **g** Cell stiffness of MSCs cultured on the three topographies was measured using nanoindentation and showed reduced stiffness on the SQ topography and increased stiffness on NSQ. Numbers in brackets represent the number of individual measurements. **h** MSCs were cultured on flat nanotopographies for 14 days in the presence or absence of the ROCK inhibitor, Y27632. Actin cytoskeleton changes were revealed by phalloidin staining (n = 15 fields per group). **i** The immunomodulatory capacity of cells cultured on flat topographies for 7 or 14 days in the presence or absence of Y27632 was assessed (n = 3 topographies per group, mean ± S.D.). **j** Fold change in proliferation index to untreated controls of MSCs grown on flat topographies in the presence of Y27632 for 7 or 14 days (n = 3 independent donors; mean of n = 4 topographies per donor). Means ± SEM and number of donors (N) are shown for each condition. ***p ≤ 0.0001; *p < 0.05; n.s., non-significant. Direct comparisons by two-tailed student T-test (Mann–Whitney) and in (**f**, **g** and **j**) by one-way ANOVA with Kruskal–Wallis test of multiple comparisons. Source data are provided as a Source data file.

## Untargeted metabolomics points to respiration as a central mechanism

It has been previously reported that metabolite depletions unique to either the chondrogenic or osteogenic differentiation of MSCs can be identified by mass spectrometry[26]. Importantly, it has been shown that these metabolites could, by themselves, induce targeted differentiation[26]. We have developed this methodology to identify metabolites involved in MSC immunomodulation. However, a key challenge with studying basic MSC mechanisms and with elucidating MSC fate and function, is that MSC populations are highly heterogenous[27]. Thus, here, we use both an enriched skeletal (Stro-1⁺) MSC population[28] and an unselected (total) commercial bone marrow-derived skeletal MSC population.

Total and Stro-1⁺ MSCs were cultured on SQ and on flat controls, and their metabolites were isolated at days 7 and 28 of culture, for mass spectrometry analysis. We annotated over 200 metabolites that changed in abundance between both MSC populations grown on SQ vs flat surfaces (Fig. 2a and Fig. S6). To provide focus, we selected only metabolites classified as true identifications (class I according to the Metabolite Standards Initiative guidelines[29]), providing 18 hits common to both time points (Fig. 2b). Of these, four metabolites were depleted at both days 7 and 28 (adenine, citrate, L-glutamic acid and niacinamide) (Fig. 2c, d), which are all involved in cellular respiration (Fig. 2e).

There is a growing body of literature on pluripotent stem cells, which demonstrates that the Warburg effect[30] supports the maintenance of their pluripotency[31,32]. The Warburg effect is a mechanism employed by cancer cells that involves a shift from oxidative phosphorylation in the mitochondria to oxidative glycolysis in the cytoplasm[30]. This metabolic shift is perhaps counterintuitive in fast-growing cancer and stem cells, as oxidative glycolysis is less efficient than oxidative phosphorylation is, in terms of adenosine triphosphate (ATP) production, which is the main source of cellular energy. However, this is mainly a problem for cells when glucose is deficient; in the body and in cell culture, glucose is in ready supply[33].

Proliferating cells, in fact, might benefit from oxidative glycolysis by maintaining carbon rather than by releasing it as $CO_2$, and from the increased metabolic supply of niacinamide adenine dinucleotide (NADH) and niacinamide adenine dinucleotide phosphate (NADPH). Both carbon supply and NADH/NADPH activity are important for amino acid, lipid and nucleotide biosynthesis, all of which are essential for generating new cells[33,34]. It is thus notable that the metabolites we have identified—adenine, citrate, L-glutamic acid and niacinamide—are all involved in cytoplasmic NADH/NADPH pathways[33,34].

To determine if mitochondrial function is reduced in the immunomodulatory MSCs cultured on the SQ nanotopography, mitochondrial activity was assayed using JC1 staining and quantified by flow cytometry (Fig. S7). This showed that mitochondrial activity was reduced in MSCs cultured on SQ after 14 days of culture, as compared to MSCs cultured on the flat control (Fig. 2f). It is noteworthy that the measurement of mitochondrial mass, using mitotracker green, and of mitochondrial function, using superoxide generation with Mitosox red, demonstrated that MSCs cultured on the SQ or flat surfaces had comparable mitochondrial physiology (Fig. 2g, h). These results suggest that in the presence of the SQ topography, mitochondrial activity is reduced but that organelle physiology itself is not affected.

We also inhibited ROCK signalling (via Y27632) in MSCs to determine if their mitochondrial activity, as measured by JC-1 expression, was affected by intracellular tension. Indeed, ROCK inhibition in MSCs cultured on flat control surfaces reduced JC-1 expression, indicating decreased mitochondrial activity (Fig. 2i) which pairs with the observation in Fig. 1f, i, j that decreased intracellular tension increases immunomodulatory phenotype.

Finally, as we are relating mitochondrial activity to intracellular tension we used image analysis of mitochondria and actin and employed Manders and overlapping coefficients to look at mitochondrial association with actin; both calculate pixel co-localisation. While MSCs on control and NSQ surfaces had similar coefficients, the co-localisation of mitochondria and actin was reduced in MSCs grown on the SQ topography and with Y27632 treatment, with the effect of the SQ nanotopography being more subtle than Y27632 treatment (Fig. S8)[35].

## Flux of heavy glucose shows increased oxidative glycolysis

To investigate changes in cellular respiration in MSCs grown on different nanotopographies, we used mass spectrometry to follow the breakdown and conversion of ¹³C-labelled glucose. Cells were cultured on SQ and flat control surfaces for 11 days in standard media followed by 3 days with ¹³C glucose-containing media. As illustrated in Fig. 3a, increased oxidative glycolysis results in increased lactate production. And indeed, MSCs cultured on SQ surfaces displayed increased glucose consumption and increased lactate production, relative to MSCs cultured on flat surfaces, while the mitochondrial tricarboxylic acid (TCA) cycle remained similar in MSCs cultured on either surface (Fig. 3b). Measurements of fluorescently labelled glucose uptake (Fig. 3c) and of extracellular lactate (Fig. 3d) confirmed the mass spectrometry data.

Previous reports have noted that lactate secreted from umbilical cord-derived MSCs can have effects on immune cells[35], and exposure to lactate can both alter MSC gene expression[36] and maintain their colony-forming potential[37]. Given the increase in extracellular lactate observed, we tested the effects of exposure to 5 mM lactate during culture on MSCs seeded on flat control materials. Adding exogenous lactate to cell culture media, in addition to that secreted by the MSCs, increased cell stiffness but did not significantly alter the ability of MSCs to suppress T cell proliferation (Fig. S9). For our bone marrow-derived MSCs, we could postulate, from the increase in cell stiffness, that the addition of this exogenous lactate may be inducing differentiation.

Together, these results demonstrate that MSCs cultured on SQ, which display an enhanced immunomodulatory capability, subtly

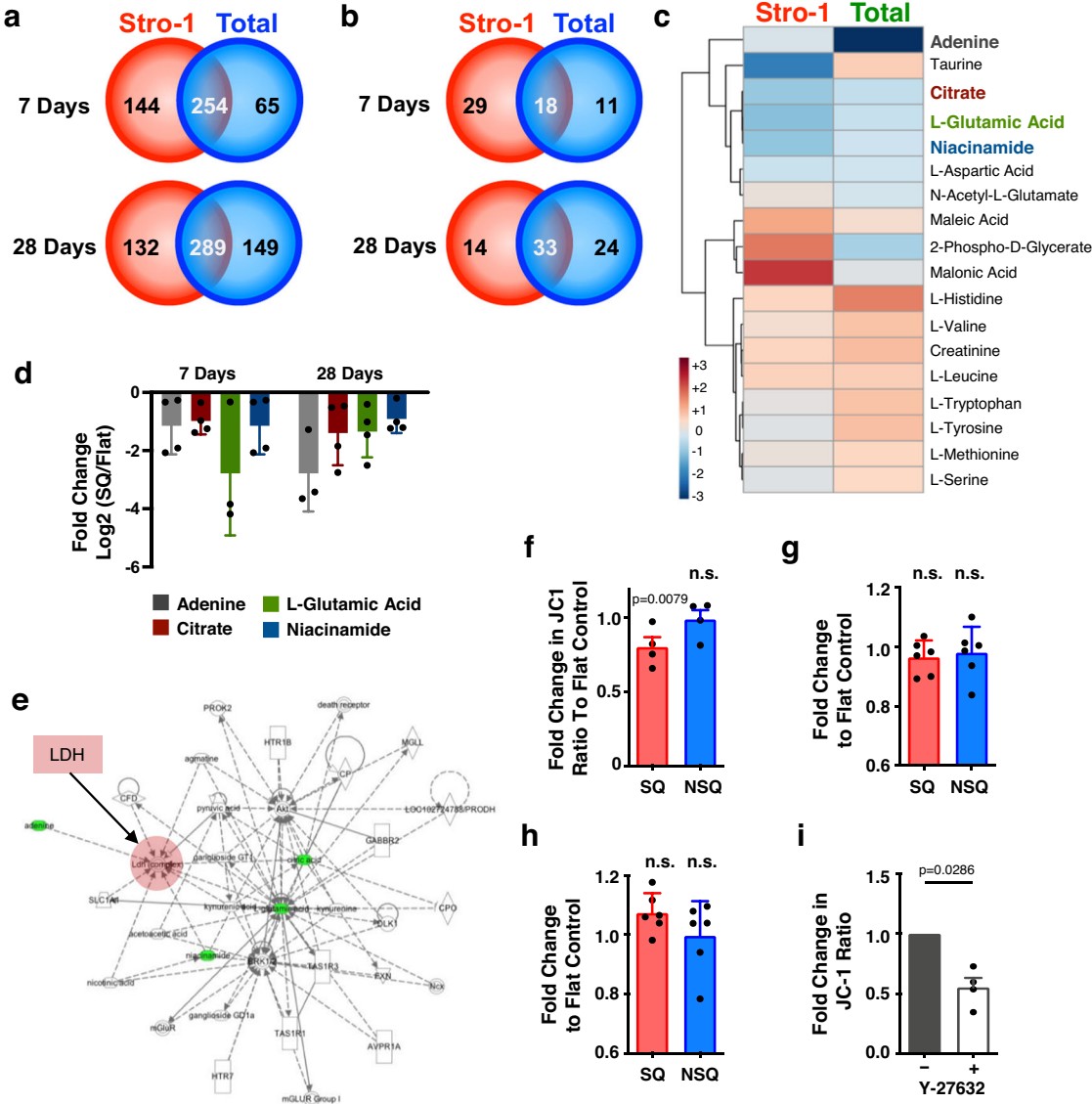

**Fig. 2 | Metabolome analysis reveals nanotopography-mediated changes in cellular respiration, independent of mitochondrial dynamics. a** Stro-1⁺ (red) or total BM (green) MSCs were cultured on SQ or flat surfaces for 7 or 28 days, and the number of metabolites specific or common to both cell types were enumerated. Common metabolites with a confidence value of 10 were identified using IDEOM software in both MSC populations (**b**), and the heat map shows the distribution of these at day 7 of culture (**c**). **d** Fold change in selected metabolite concentrations. Data in (**a**–**d**) show mean of 6 nanotopographies per condition. **e** Biochemical network analysis of metabolite changes in MSCs cultured on flat versus SQ. **f** Changes in mitochondrial activity were measured using JC-1 staining, in MSCs cultured on SQ or NSQ nanotopographies relative to flat control. Each data point is the mean of 3 technical repeats per independent donor ($n = 4$). Comparisons by one-way ANOVA (*$p = 0.0079$). **g**, **h** Mitochondrial mass (Mitotracker Green) and superoxide generation (Mitosox Red) were also evaluated by flow cytometry (single measurement from $n = 4$ donors). **i** Stro-1⁺ MSCs were cultured on flat and SQ nanotopographies ($n = 4$ topographies per donor, $n = 4$ independent donors) for 14 days in the presence (hatched bars) or absence (open bars) of the ROCK inhibitor, Y27632, and changes in mitochondrial activity measured using JC-1 staining. Means ± SEM are shown in **f**, **h** and **i** and number of donors (N) are shown for each condition. Means ± S.D. in **d**, **g** and **h** by one-way ANOVA. Direct comparisons of means by two-tailed student T-test (Mann–Whitney), *$p < 0.05$; n.s., non-significant. Source data are provided as a Source data file.

increase oxidative glycolysis, as is typically observed during the Warburg effect, compared to MSCs cultured on the flat control.

No difference in glucose uptake, nor in oxidative glycolysis, was observed in MSCs cultured on the NSQ surface with ¹³C-labelled glucose, as compared to controls, but oxidative phosphorylation was increased, as indicated by enhanced L-glutamic acid production (Fig. S10). This finding supports the prevailing view that differentiation is energetically demanding for stem cells, and thus causes increased ATP production through increased oxidative phosphorylation[38–41]. It is further noteworthy that osteoblasts derived from MSCs using the NSQ surface, are slow growing cells[24,42,43]. This highlights the requirement of oxidative phosphorylation for stem cell expansion.

## MSC immunomodulatory capacity is enhanced by decoupling oxidative phosphorylation

The central aim of this study was to demonstrate that nanotopography can be used to identify pathways that can be exploited to maintain the immunomodulatory capabilities of MSCs in culture for a prolonged period of time. Our metabolomic analysis indicates that increased oxidative glycolysis could be key to achieving this aim. We hypothesized that by decoupling mitochondrial activity, to force cellular respiration to shift from oxidative phosphorylation to oxidative glycolysis, we could promote the MSC suppressive phenotype. To test this, we used 2,4-dinitrophenol (DNP), an ionophore that dissipates proton gradients across mitochondria, preventing

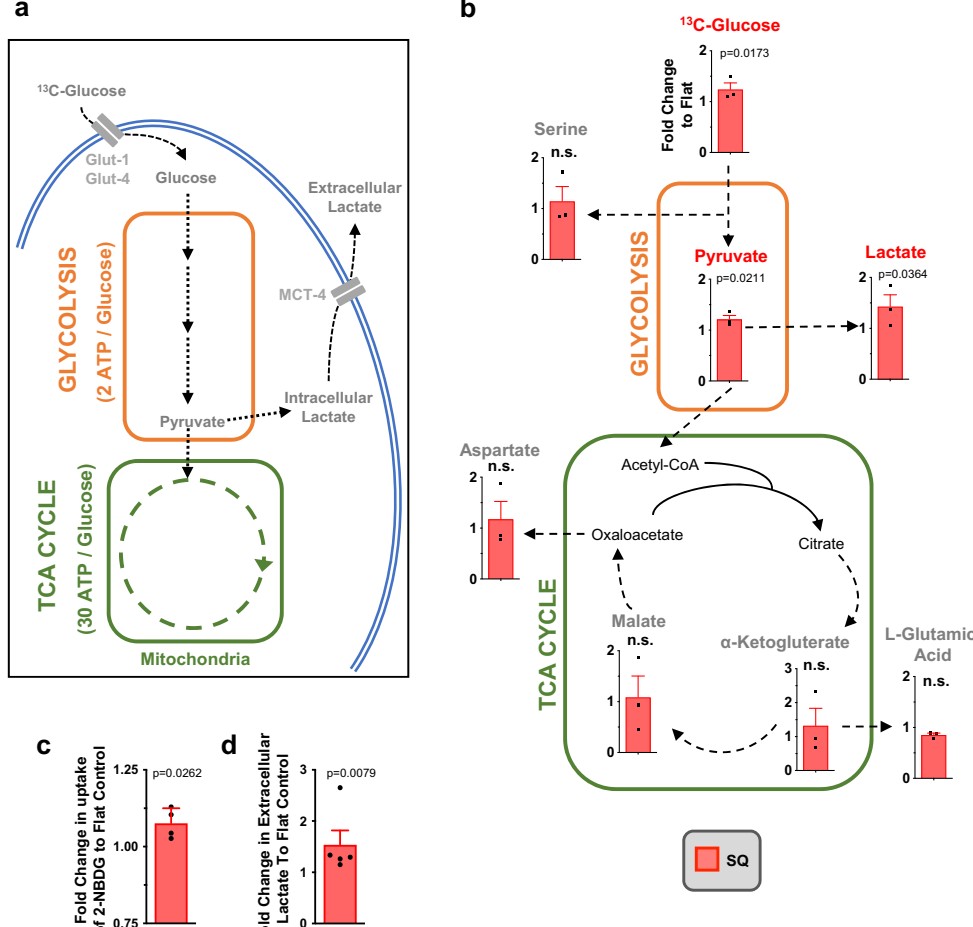

**Fig. 3 | MSCs increase oxidative glycolysis on SQ nanotopographies, as revealed by [$^{13}C_6$]-glucose tracing. a** Schematic of MSC respiration during culture. Changes to these pathways relative to MSCs on planar controls can be traced using heavy labelled [$^{13}C_6$]-glucose. **b** Stro-1+ MSCs were cultured for 14 days on nanotopographies in the presence of [$^{13}C_6$]-glucose for 72 h. LC-MS was then used to measure the conversion and abundance of [$^{13}C_6$]-labelled metabolites in the glycolysis and TCA cycle pathways. Graphs show a fold change in [$^{13}C_6$]-labelled metabolites in MSCs cultured on SQ relative to flat nanotopographies. **c** MSCs were cultured for 14 days on SQ or flat surfaces ($n = 3$-4 topographies per donor, $n = 4$ independent donors), and glucose uptake was measured using 2-NBDG (a fluorescent glucose analogue) by flow cytometry. **d** Cell culture supernatants were collected from MSCs grown for 14 days on flat or SQ nanotopographies, and extracellular secreted lactate was quantified ($n = 2$ samples per donor, $n = 3$ independent donors). Means ± SEM and number of donors (N) are shown for each condition (Direct comparisons by two-tailed student T-test (Mann–Whitney), *$p < 0.05$; n.s., non-significant). Source data are provided as a Source data file.

the proton motive force that produces ATP-related energy thus driving glycolysis[44]. MSCs were cultured on flat surfaces in the presence or absence of 0.5 mM DNP. We observed an increase in the immunomodulatory phenotype of DNP-treated MSCs relative to untreated controls (Fig. 4a). One mechanism of MSC-mediated immunomodulation is the upregulation and release of indoleamine 2,3-dioxygenase (IDO), following MSC exposure to interferon-gamma (IFNγ), which is produced by activated T cells. IDO expression limits T cell proliferation through the degradation of extracellular tryptophan (a key amino acid required by T cells during proliferation), and is thus associated with immunomodulation[45,46]. After priming MSCs with 100 ng/ml IFNγ overnight, we observed a similar upregulation of *IDO1* expression in both control and DNP-treated MSC populations (Fig. 4b). This demonstrates that DNP-treated MSCs which are utilising oxidative glycolysis for energy production can still respond to exogenous T cell stimuli in the form of IFNγ.

The addition of DNP also promoted the retention of MSC markers, CD44, CD90 and CD166[47] after 14 days in culture over that seen in untreated control cells (Fig. 4c). This shows that MSCs undergoing oxidative glycolysis retain markers of multipotency, as well as their immunomodulatory phenotype.

## Bioactive metabolites can enhance immunomodulatory capacity and growth

From our results, we hypothesised that the addition of individual metabolites identified in Fig. 2a–d, namely adenine, citrate, niacinamide and L-glutamic acid, that were seen to link to respiration (Fig. 2e), might stimulate the MSC immunomodulatory phenotype. Indeed, when MSCs were treated with either adenine or L-glutamic acid, this significantly increased immunomodulation (adenine $p = 0.0079$; L-glutamic acid $p = 0.0079$) as revealed by co-culture with CFSE-labelled T cells (Fig. 5a). This agrees with our emerging hypothesis that MSC phenotype is related to the need to build new cells. Adenine is a purine nucleobase used for DNA synthesis in proliferating cells and is also a component of DNA and RNA and it also forms part of the NADH/NADPH dinucleotide (as well as ATP and flavin adenosine dinucleotide (FAD)). L-glutamic acid is an amino acid used in protein production. Thus, both directly fuel production of new cell components.

Non-essential amino acid anabolism can be split into three categories[47,48]. In the first two categories, amino acid biosynthesis is TCA-cycle independent, with amino acids being derived from the pentose phosphate pathway and from glycolysis. Amino acids generated in this way include alanine, histidine, isoleucine, leucine, phenylalanine, serine, tryptophan, tyrosine and valine[47,48]. The third category

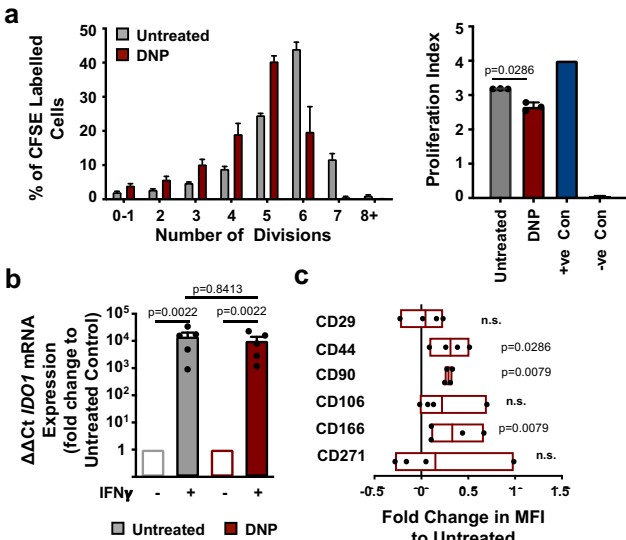

**Fig. 4 | Uncoupling oxidative phosphorylation increases MSC immunomodulation. a** Stro-1[+] MSCs were cultured in the presence or absence of DNP for 14 days, then co-cultured with CFSE-labelled, IL-2 and PHA stimulated PBMCs for a further 5 days. Proliferation was assessed by flow cytometry. Data in left graph is a representative experiment ($n = 4$ replicates per group; mean ± S.D.); data in right graph shows the proliferation index of 3 independent experiments ($n = 3$; mean ± SEM). **b** Following culture with or without DNP, MSCs were challenged with IFN-γ for 24 h and IDO1 expression was evaluated by qPCR ($n = 1$ sample per donor, $n = 5$ independent donors). **c** Fold change in MSC surface marker expression following DNP treatment, relative to untreated controls, as assessed by flow cytometry (at least 5000 cells per sample from $n = 4$ independent donors). Floating box plots show mean with high and low values (CD29, min −0.227 max 0.2319 mean 0.0446; CD44, min 0.0853 max 0.51276 mean 0.3106; CD90, min 0.2485 max 0.3313 mean 0.2906; CD106, min −0.0182 max 0.6987 mean 0.2195; CD166 min 0.1042 max 0.6667 mean 0.2906; CD271 min −0.2814 max .0987 mean 0.14892). Means ± SEM and number of donors (N) are shown for each condition. Multiple comparisons (**a** and **b**) by two-way ANOVA with Tukey's multiple comparison test or direct comparisons by two-tailed student T-test (Mann–Whitney) in (**c**). \*\*\*$p \leq 0.0001$; \*$p < 0.05$; n.s., non-significant. Source data are provided as a Source data file.

consists of TCA cycle-derived amino acids, such as lysine, methionine and threonine, which are derived from canonical aspartate, and also arginine, glutamine and proline, which are derived from canonical glutamic acid (glutamate)[47,48]. Analysis of our untargeted metabolomic data from the MSCs at days 7 and 28 showed that the canonical TCA cycle-derived amino acids, L-glutamic acid and L-aspartate, are depleted in the MSCs, even at day 28 (Fig. 5b). We therefore propose that in the immunomodulatory MSCs, L-glutamic acid and L-aspartate become depleted because the TCA cycle does not increase in balance with oxidative glycolysis. This is interesting as many of these amino acids are considered to be conditionally essential and at times of high growth are required from diet[47,48]. Thus, we propose that as MSCs rapidly grow and employ glycolysis, they become depleted of these amino acids. It is also noteworthy that at days 7 and 14 of culture, the addition of niacinamide significantly increased Stro-1[+] MSC growth (Fig. 5c). This again, illustrates that the identified metabolites are involved in new cell production.

In order to assess if the addition of the selected metabolites altered Stro-1[+] MSC respiration directly, mitochondrial function was measured using Seahorse flux analysis. Cells treated with metabolites were replated and the Cell Mito Stress test was performed using the Seahorse XFe24 analyser. All treatments showed similar mitochondrial function as determined by oxygen consumption rate (OCR) (Fig. 5d). However, treatment with adenine or niacinamide increased baseline measurements of glycolysis through extracellular acidification rate (ECAR) compared to untreated controls (Fig. 5d–f; energy

map shown in Fig. S11). This supports our hypothesis that increasing glycolysis through the addition of these metabolites promotes the naive immunomodulatory functions of MSCs. This data allows us to postulate that cellular respiration links to cellular tension which influences MSCs immunosuppressive potential. In order to investigate this concept, Stro-1[+] MSCs were cultured in the presence of metabolites identified in Fig. 2 for 14 days or Y27632 for 7 days. Western blot analysis of phospho-myosin revealed that addition of exogenous niacinamide acts to reduce intracellular tension in MSCs, comparable to addition of Y27632 (niacinamide $p = 0.0212$; Y27632 $p = 0.0205$) (Fig. 5h).

Treatment of MSCs with these exogenous bioactive metabolites has the potential to maintain a naive phenotype in standard 'normoxic' high oxygen culture conditions. This observation is further supported, as treated cells retain the ability to differentiate into osteoblastic and adipogenic lineages when instructed, a key defining functional feature of naive MSCs[20] (Fig. S12). However, despite the promotion of T-cell immunosuppression and lineage differentiation other functional immunomodulatory readouts to assess the therapeutic potential of these cells was employed. First, considering apoptosis, levels of caspase 3&6 were measured for MSCs cultured with mixed metabolites (adenine, niacinamide, glutamic acid and citrate), with lactate and with Y-27632 to lower intracellular tension; increased caspase levels compared to control culture was noted (Fig. S13a). When MSCs were treated with metabolites and subsequently co-cultured with pre-activated PBMCs and there was no difference in susceptibility to apoptosis in any of the treated cells compared to controls (Fig. S13b). It should be noted that treatment with our metabolite mix or lactate also increases caspase 3&6 expression, again mimicking the effects of the nanotopographical growth surface (Fig. S3). This, again, suggests that metabolite priming MSCs to promote immunomodulation activities retains their ability to respond to activated T cells and induce apoptosis in agreement with studies showing that this MSC physiological effect is correlated to potential therapeutic efficacy[20,21].

Functional, immunomodulatory MSCs have a number of strategies to dampen an immune response including the release of paracrine soluble factors (e.g. transforming growth factor (TGFβ), hepatocyte growth factor (HGF), prostaglandin E2 (PGE2), prostaglandin-endoperoxide synthase (Cox-2), IL-6, etc.) as well as the release of enzymes, extracellular vesicles and receptor-mediated cell-cell contacts[49,50]. We, therefore, examined the secretome of metabolite-treated MSCs after priming with the pro-inflammatory mediators tumour necrosis factor (TNFα) and interferon gamma (IFNγ)[51]. Firstly, we observed that secretion of IDO1 is the main immunomodulator of T cell suppression in our co-culture (Fig. S14). Blocking with the established IDO1 inhibitor 1-MT reduced the ability of MSCs to suppress proliferation. Blocking other established immunomodulatory pathways (Cox-2, PGE-2, NO)[51] had little impact on T cell suppressive properties (Fig. S14a). Secondly, we performed the T cell suppression assay with MSCs treated with both the mixed metabolites in the presence of 1-MT mediated IDO1 inhibition. Here, the metabolite-enhanced MSC immunomodulatory effect became negligible, again showing that the metabolite-driven T cell suppression is mainly mediated through IDO1 (Fig. S14b).

We then investigated the release of soluble factors after priming. Secretion of TGF-β, PGE2, IL-8 was not significantly affected after lactate or Y-27632 treatment (Fig. S15a–c). Similarly secreted IDO1 activity levels were also not affected as assessed by the accumulation of L-kynurenine (Fig. S14d). However, treatment with the metabolite mixture significantly increased secretion of PGE2 and IL-8 compared to untreated cells, supporting the enhanced immunomodulatory potential of these cells (Fig. S15b, c). In addition, induction of TNF-stimulated gene 6 protein (TSG-6), a TNF responsive secreted protein released by MSCs was assessed. It has both anti-inflammatory effects on macrophages and neutrophils as well as impacting on the

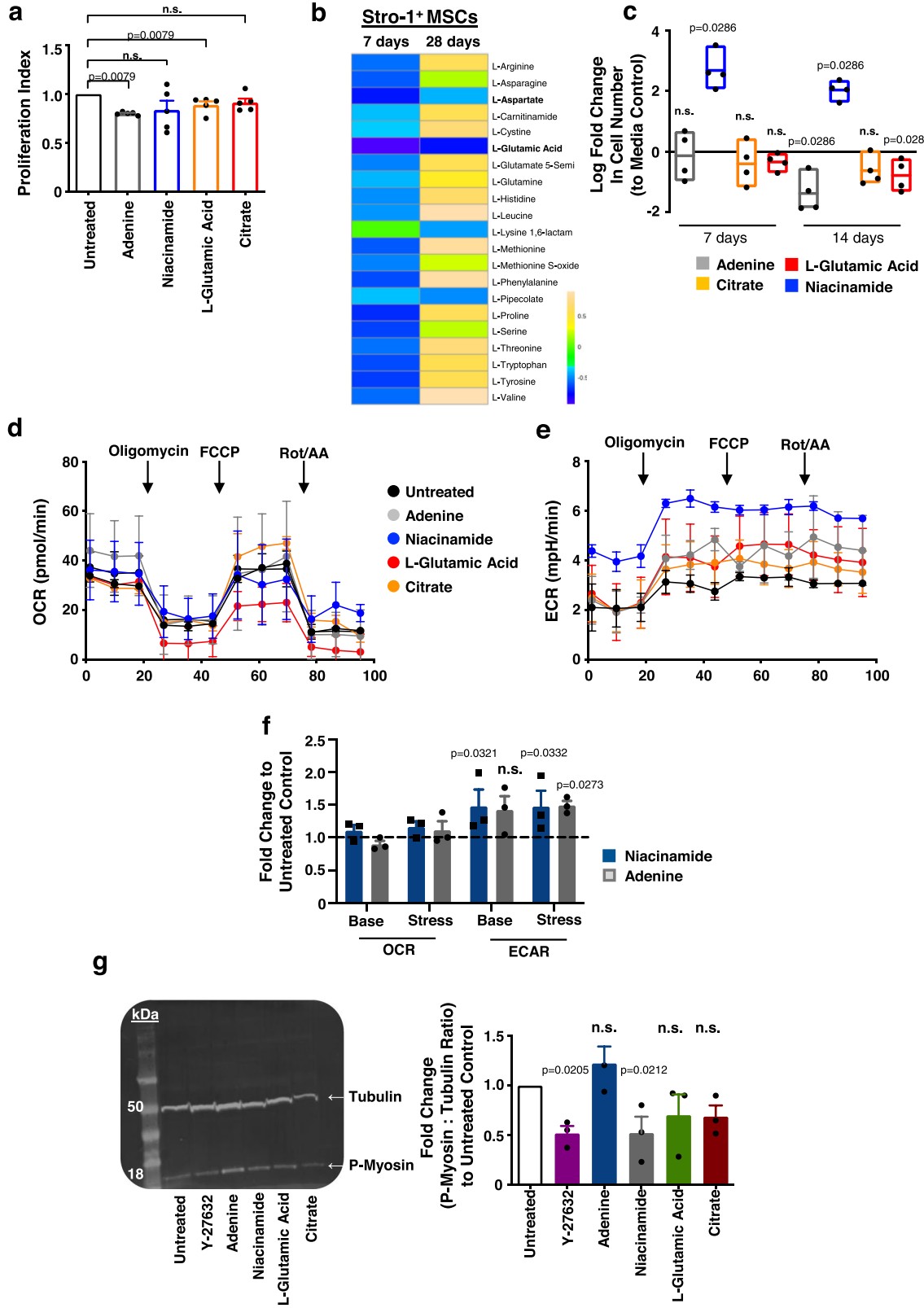

presentation of chemokines with surrounding matrix proteins, thus is an important factor in resolving inflammatory environments[52]. Furthermore, recent in vivo work demonstrates this factor helps to maintain the stemness of MSCs[53]. After 24 h of priming, the mixed metabolite-treated MSCs maintained their ability to induce TSG-6 when exposed to TNF-α (Fig. S15e, f). This further demonstrates the ability of metabolite treatment to retain key physiological functions

of MSCs correlated to their naivety, stemness and immunomodulatory potential.

To demonstrate the direct functional activity of treated MSCs on other immune cells, the ability to promote regulatory T cell (TReg) programming during co-culture with T cells was also evaluated; this is a key immunomodulatory function of naive MSCs (Fig. S16). Metabolite-treated cells were able to support TReg differentiation, increasing the

**Fig. 5 | Addition of defined metabolites influences MSCs immunomodulation capability. a** Stro-1+ MSCs were cultured in the presence of metabolites for 14 days, followed by co-culture with CFSE labelled, IL-2 and PHA stimulated PBMCs for a further 5 days. Graph shows the proliferative index of T cells normalised to untreated controls and is representative of 2 independent experiments ($n = 4$ topographies per group, mean ± S.D.). **b** Changes in amino acid synthesis in Stro-1+ MSCs grown on SQ versus flat nanotopographies for 7 or 28 days. At both time points, L-glutamic acid and L-aspartate were depleted ($n = 6$ topographies per group). **c** Stro-1+ MSCs were cultured with selected metabolites and fold change in total cell number was measured by flow cytometry relative to untreated controls cytometry (at least 5000 cells per sample from $n = 4$ independent donors). Floating bar plots show mean with high and low values at day 7 (Adenine, min −0.9783 max 0.6816 mean −0.1351; Niacinamide, min 2.0646 max 3.5192 mean 2.6851; Glutamic Acid, min −1.1782 max 0.4436 mean −0.399; Citric Acid, min −0.7059 max −0.0395 mean −0.3411) and day 14 (Adenine, min −1.8668 max −0.5363 mean −1.3839; Niacinamide, min 1.6152 max 2.3692 mean 2.0345; Glutamic Acid, min −1.0469 max

0.0458 mean −0.6289; Citric Acid, min −1.3269 max −0.2179 mean −0.7857). Representative Seahorse XF Mito Stress Test analysis of metabolite-treated Stro-1+ MSCs and normalised mitochondrial respiration shown as oxygen consumption rate (OCR) (**d**) and extracellular acidification rate (ECAR) (**e**). Data in **e** and **f** is from $n = 3$ technical repeats from one representative donor. **f** Fold change of baseline (Base) and stressed (Stress) OCR and ECAR in metabolite-treated Stro-1+ MSCs compared to untreated controls (mean of 3 technical replicates per donor, $n = 3$ independent donors). **g** Stro-1+ MSCs were cultured with selected metabolites for 14 days or ROCK inhibitor Y-27632 for 7 days. Levels of phospho-myosin (18 kDa) relative to β-tubulin (50 kDa) was quantified by western blotting. Blot is representative of 3 independent donors. Graph on right shows quantitative changes in phospho-myosin expression normalised to β-tubulin. Means ± SEM and number of donors (N) are shown for each condition. Comparisons (**a, g**) by two-way ANOVA with Dunnett's multiple comparison test or two-tailed student T-test (Mann–Whitney) in (**c, f**). *$p < 0.05$; n.s., non-significant. Source data are provided as a Source data file.

number of these cells in co-culture compared to untreated MSCs. These data suggest that treatment with metabolites will not impair their therapeutic potential during expansion in culture.

### Large scale growth of immunomodulatory MSCs

For cell therapies, large numbers of MSCs are required ($\sim$1–2$\times 10^6$/kg per treatment) with a median dose of 100$\times 10^6$ cells intravenously[54]. MSC manufacture for cell therapies can be from autologous or allogeneic donation. Autologous MSC therapies need to be derived from the patient and will be dependent on how their cells expand. Allogeneic approaches, however, have clinical advantages in that cell banks can be produced from highly proliferative donor lines and then therapies can be pre-manufactured and stored frozen[50].

To see if our metabolites allow expansion of MSCs at a clinical scale with enhanced immunomodulatory phenotype we took an autologous-analogous approach using 5-layer cell stacks (culture area of 3180 cm², Fig. 6a) to test three metabolite treatment approaches. Firstly, all four identified metabolites were used together (Fig. 2d, mixed); while the cells maintained immunomodulatory phenotype, cell growth was slowed (Fig. 6b, c). Next, we trialled niacinamide-containing media alone as it had given good MSC growth characteristics (Fig. 5c). However, while enhanced growth was seen, immunomodulatory phenotype, while lower than control conditions, was more variable (Fig. 6b, c). This was comparable to previous findings of donor response to niacinamide in smaller culture settings (Fig. 5a). Finally, we tested MSCs treated with niacinamide-containing media for 7 days and then switched to mixed metabolite media. Here, enhanced growth and consistent immunomodulation were observed (Fig. 6b, c).

As proliferation index does not include T cells that have not divided and as the T cell population is heterogeneous with cells undergoing different numbers of divisions, small changes in proliferation index can indicate large difference in immunomodulatory phenotype. Looking at cytometry data (Fig. 6d), it was observed that untreated MSCs supported a T cell profile similar to positive control, i.e. they were barely immunomodulatory after cell stack expansion. However, the niacinamide then mixed metabolite-treated MSCs supported T cell populations comprising highly immunomodulatory cells (similar to negative control) through to less immunomodulatory cells (Fig. 6d). This data is critical as it shows that we have achieved faster expansion, to the point we are manufacturing clinical doses, of more immunomodulatory MSCs. Importantly, these expanded cells still retain their susceptibility to apoptosis when cultured with activated PBMCs, suggesting they retain therapeutic potential using this correlate of function (Fig. 6e). Looking at MSC markers, enhanced expression of naive associated markers (CD166, CD271) was noted in the niacinamide then mixed metabolite cultures at end of expansion (Fig. 6f).

## Discussion

In this study, we set out to investigate whether nanotopography can enhance the immunomodulatory capacity of MSCs through increased oxidative glycolysis. A growing number of reports indicate that oxidative phosphorylation increases in MSCs, as well as in other stem cells, including pluripotent[39] and hair follicle[40] stem cells, as they undergo differentiation[41,51,52], and that MSCs in standard culture are more glycolytic[41,51,52]. However, when MSCs are cultured on standard flat tissue culture plastics, they undergo phenotypical drift and lose certain characteristics, such as their immunomodulatory capacity, as shown in Fig. 1c, d. In this study, we use the SQ nanotopography to regulate MSCs and to maintain their naive multipotent cell phenotype for prolonged periods of in vitro culture (Fig. 1e). This allows us to definitively show that shifting MSC respiration towards oxidative glycolysis is key to maintaining their immunomodulatory phenotype. Our data thus highlight the importance of properly controlling MSC experiments to correctly interpret data on cell function.

At the extremes of intracellular tension with adipocytes (low cytoskeletal tension) and osteoblasts (high cytoskeletal tension), it is noteworthy that these cells are slow growing, either through lack of response to mitogens due to lack of adhesion (adipocytes) or have negative feedback loops stimulated due to high adhesion (osteoblasts)[24,53–55]. The changes in morphology, intracellular tension and cell biochemistry between MSCs and fibroblasts, however, are much more subtle[2,3,24]. Indeed, MSCs were first identified as precursors of mechanocytes[56], now known as fibroblasts, and as fibroblast colony-forming units[57], due to the similarity in their appearance and their ability to form stromal tissues. These subtle differences, however, are important and provide a basis for understanding MSC behaviour in culture. Critically, for understanding maintenance of MSC phenotype (which has regenerative and therapeutic potential) versus phenotypic drift towards a fibroblast-like, committed, state[24]. MSCs have lowered intracellular tension compared to fibroblasts[58], but differ to the almost complete loss of intracellular tension noted with adipocytes[2]. In this study, we demonstrate that their suppressive, immunomodulatory ability depends on this lowered cytoskeletal tension and implicate a relationship between the contractile cytoskeleton and mitochondria. Interestingly, as both MSCs and fibroblasts are 'intermediate tension' phenotypes they are both faster growing phenotypes compared to committed mature stromal structural cells.

In vivo, in the bone marrow niche, hypoxia is likely to help to maintain enhanced glycolysis in MSCs via hypoxia inducible factor 1 (HIF-1α)[59,60]. HIF-1α promotes the expression of pyruvate dehydrogenase kinase, which prevents pyruvate from entering the TCA cycle, thereby inhibiting mitochondrial respiration[59,60]. Our analysis of RNAseq data, obtained from Stro-1+ MSCs cultured on SQ vs flat nanotopographies after 24 h of culture, showed oxidative phosphorylation to be significantly repressed, but only a limited regulation of

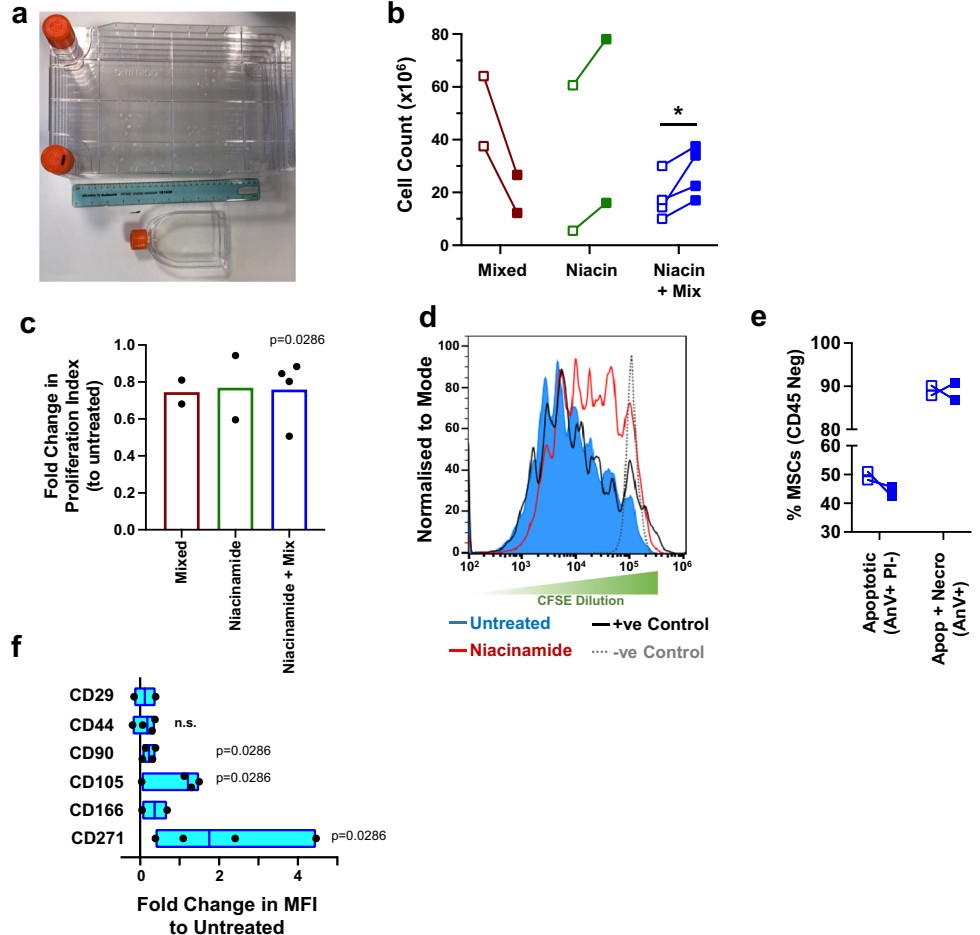

**Fig. 6 | Upscaling cell cultures using metabolite treatment supports production of immunosuppressive MSCs. a** Stro-1+ MSCs were grown using five-layer cell stacks to increase cell yields (3180 cm[2] growth area, shown next to a standard T75 flask). **b** Numbers of cells recovered from the cell stacks after two weeks of culture with (filled square) and without (empty square) metabolite treatment. Treatments were mixed (adenine, niacinamide, L-glutamic acid and citrate), niacinamide only (niacin) and niacinamide followed by mixed. **c** MSCs grown in large culture conditions were harvested, replated and their immunosuppressive capacity measured co-cultured with CFSE-labelled, IL-2 and PHA stimulated PBMCs for a further 5 days. Proliferation was assessed by flow cytometry. Fold change in the proliferation index to matched untreated controls shown for each metabolite treatment.
**d** Representative flow cytometry histogram of CFSE dilution in T cells co-cultured with niacinamide and mixed treated MSCs for 5 days. **e** Stro-1+ MSCs were cultured with niacinamide followed by the four metabolite mixture and assessed for

susceptibility to apoptosis when co-cultured with activated PBMCs at 1:10 ratio using Annexin-V and PI staining by flow cytometry (*n* = 2 independent donors). The proportion of apoptotic (Annexin-V+PI) and necrotic (Annexin-V+PI+) are shown for either untreated (empty squares) or metabolite-treated (filled squares) MSCs. **f** Fold change in MSC surface marker expression following niacinamide and mixed treatment for 14 days, relative to untreated controls, as assessed by flow cytometry (at least 5000 cells per sample from *n* = 4 independent donors). Floating box plots show mean with high and low values (CD29, min 0.8442 max 1.3912 mean 1.1177; CD44, min 0.8056 max 1.3757 mean 1.1353; CD90, min 1.0511 max 1.39 mean 1.2236; CD105, min 1.0342 max 2.494 mean 1.9871; CD166, min 1.0481 max 1.3575 mean 1.9871; CD271, min 1.3867 max 5.460 mean 3.0874). All graphs show *n* = 2–4 donors for each metabolite treatment. Paired comparisons (**b**, **c** and **f**) by two-tailed student T-test (Mann–Whitney) *p < 0.05; n.s., non-significant. Source data are provided as a Source data file.

HIF1α signalling was observed (Fig. S17). This supports the hypothesis that in normoxia, on the SQ nanotopography, oxidative glycolysis is activated in MSCs through changes in cytoskeletal tension rather than via a hypoxic mechanism.

We propose that the tension-based mechanism that activates oxidative glycolysis can be exploited to support MSC growth in standard culture conditions, through the addition of specific metabolites that support cell growth, namely adenosine, L-glutamic acid, citrate and niacinamide. We propose that these metabolites are required for MSC growth, and for the maintenance of the MSC phenotype, by providing the precursors of anabolic co-factors and conditionally essential amino acids. Their provision could thus support enhanced MSC growth in the absence of oxidative phosphorylation, which is important because oxidative phosphorylation is associated with MSC differentiation[41,51,52]. It is interesting to note that while maintained MSCs on SQ grew at the same rate as those in control cultures (Fig. 1e),

optimising with metabolites allowed for faster growth of the MSCs (Fig. 6b).

The implications of these findings are important for the cell therapy field because they could be applied to support the large-scale expansion of MSCs for therapeutic application; indeed we demonstrate the potential to manufacture enhanced clinical-scale MSC doses. Our results also demonstrate the value of material-based tools for understanding stromal cell progenitor function and fate, and of their broader potential application across other biological systems and tissues. A schematic of key findings is provided in Fig. 7.

## Methods
### Materials fabrication
Polycarbonate nanotopographical culture surfaces comprised of 120 nm diameter pits were produced by injection molding using an Engle Victory 28 hydraulic injection moulder. Master SQ (300 nm

a

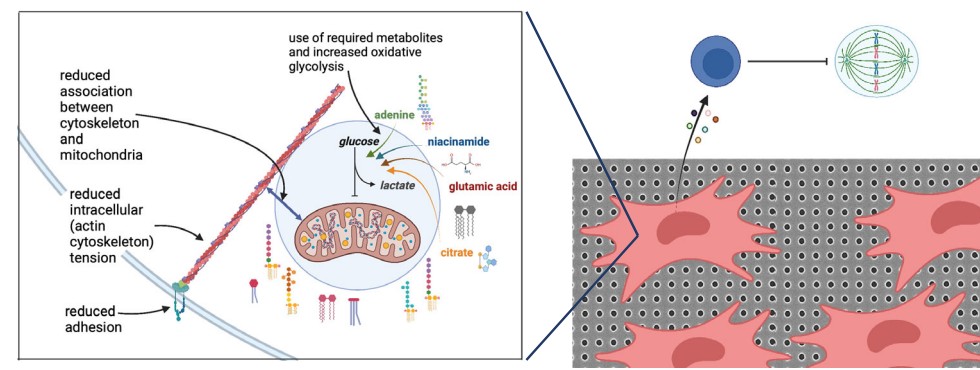

b

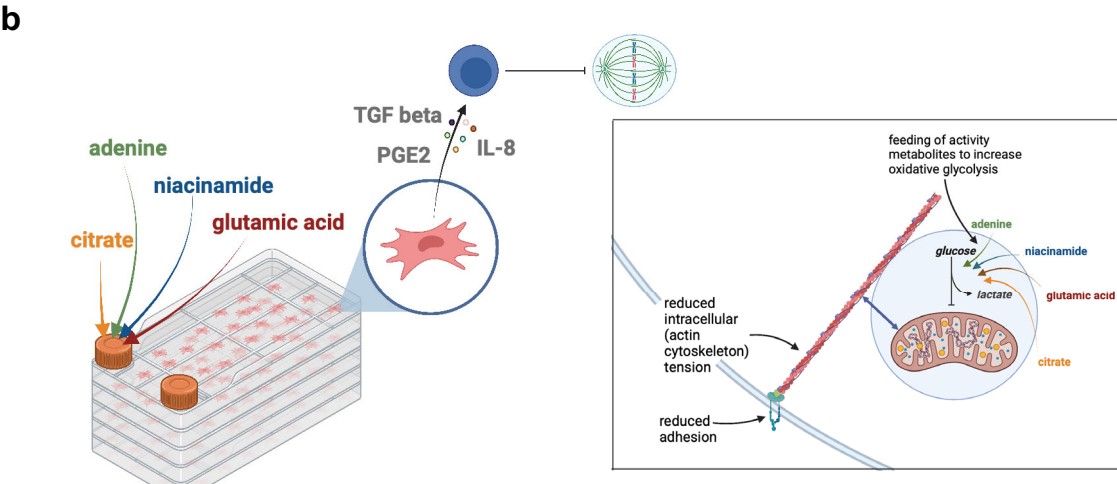

**Fig. 7 | Schematic illustrating main findings. a** Culture on the SQ nanotopography causes a lowering of cell adhesion, cytoskeletal tension and mitochondrial association with actin microfilaments. This results in use of adenine, niacinamide, glutamic acid and citrate by the cells to drive glycolysis resulting in reduced T cell proliferation. **b** Addition of the glycolysis-driving metabolites into large-scale MSC culture results in similar adhesion and intracellular tension reduction as seen when the cells are cultured on the SQ nanotopography along with secretion of TGFβ, PGE-2 and IL-8 and resultant reduced T cell proliferation. Red cells = MSCs, blue cells = T cells. Created with BioRender.com.

centre-centre spacing and 100 nm depth arrays) or NSQ (arrays with an additional 50 nm offset spacing of pits) nickel dies were produced from patterned resists of silicon-coated substrates fabricated on silicon-coated 100 nm PMMA (Elvacite 2041, Lucite International) by electron beam lithography. The silicon substrates were exposed to a 50 kV electron beam, developed in 1:3 MIBK:IPA for 30 s, rinsed in IPA and dried in a nitrogen stream. Patterned resists were sputtered coated ith 50 nm Ni-V and electroplated to a thickness of ~300 μm (outsourced to DVDNorden, Denmark). Nickel shims were chloroform cleaned for 10–15 min in an ultrasonic bath, rinsed in acetone and IPA, and dried once more in gaseous nitrogen. Surfaces were produced by injection moulding through heating polycarbonate (Makrolon® OD2015) to 180 °C and applying a clamping force of 250 kN to imprint nanotopographical pattern onto the surface of the polycarbonate. Final dimensions of each substrate being 24 mm × 24 mm. After cooling to 70 °C, separation of the press and polymer surfaces was achieved. Unpatterned (flat) polycarbonate substrates were injection moulded against planar shims and used as controls.

### Contact angle measurement and surface energy analysis

Static contact angles between liquid droplets and material surfaces of interest, including flat, SQ, and NSQ surfaces were measured by dropping 3 μL drop of test liquids on the surfaces using a Theta optical tensiometer (Biolin Scientific, Stockholm, Sweden). The test liquids used include HPLC grade water, diethylene glycol (Sigma-Aldrich, 99%), and formamide (Sigma-Aldrich, 99%). The surface tension values of the test liquids and the corresponding components used in this study are listed in Supplementary Table S1. Ten static contact angle measurements were taken for each test liquid on the material surfaces. The total surface energy was calculated using Owens-Wendt-Rabel-Kälble (OWRK) method[61].

### MSC isolation and culture

Stro-1 positive MSCs were derived from waste human bone marrow samples obtained from haematologically normal patients undergoing hip replacement surgery at Southampton General Hospital and Spire Southampton Hospital following informed consent as previously described[62]. Use of samples was approved by the University of Southampton local ethics committee (NRES number: 194/99/1, LREC number: 31875) and all methods were performed in accordance with the relevant guidelines and regulations. Consent allows age and sex to be known but not linked to patient information; Fig. S18 shows the MSC diversity to illustrate donor inclusivity. Commercial BM MSC were purchased from PromoCell (PromoCell GMBH, Germany). In all experiments, cells from multiple donors were used, as outlined in the figure legends and summarised in Supplementary Table 4. MSCs were routinely cultured in basal media (comprising DMEM supplemented with 10% (v/v) Foetal Calf Serum, 2.5 mM L-glutamine, 80 U/ml

penicillin, 0.11 mM streptomycin, 1x non-essential amino acids, 1 mM sodium pyruvate (all Sigma-Aldrich) and 0.2 µg/ml Fungizone amphotericin B (Gibco)) and used for experiments at passages 2–4. Cells were seeded on nanotopographies at 1000 cells/cm$^2$ using seeding devices as previously described[8] and routinely cultured for 14 days, or where indicated 28 or 42 days. Basal culture media was replaced every 2–3 days.

### Influencing MSC respiration in vitro

MSCs were forced to into a glycolytic state by culturing them for 14 days in the presence of 0.5 mM 2,4-Dinitrophenol (DNP), a mitochondrial decoupling protonophore. Media was replaced every 2–3 days. To assess the impact of extracellular metabolites on cell function, MSCs (1000 cells/cm$^2$) were cultured in basal media supplemented with either 5 mM adenine, 5 mM Citrate, 10 mM Niacinamide or 25 mM L-Glutamic Acid. Media was replaced every 2–3 days. To assess the effect of cytoskeletal tension on respiration, MSCs were cultured for 7 or 14 days in basal media containing 10 µM of the ROCK inhibitor Y27632 (Abcam). Media was changed every 2 days.

### Flow cytometry

To phenotype MSCs after culture, cells were washed once with PBS and detached using Accutase (ThermoFisher). Cells were stained on ice for 30–45 min using antibodies outlined in Supplementary Table S2 in flow cytometry buffer (PBS supplemented with 0.5% BSA and 0.5 mM EDTA). Cells were then washed twice with PBS and analysed using an Attune NXT flow cytometer (ThermoFisher). To identify regulatory T cells after co-culture, cells were first labelled with fixable viability dye eFluor 780 (1:2000, ThermoFisher) according to the manufacturers protocol. Cells were then incubated on ice for 30–60 min with antibodies to extracellular markers (CD45, CD4, CD8, CD25) before being fixed and permeabilised using the Tru-Nuclear fixation kit (eBioscience). Intracellular staining was performed using an anti-FoxP3-PE or isotype-PE control for 60 min at RT. Cells were washed and data collected using a Cytoflex S cytometer (Beckman Coulter).

Mitochondrial function was quantified using the membrane potential dye JC-1 (ThermoFisher). Cells were cultured in basal media containing 2 µM JC-1 for 30 min at 37 °C, then detached and analysed by flow cytometry. Cells treated with JC-1 and 50 µM carbonyl cyanide 3-chlorophenylhydrazone (CCCP, Sigma-Aldrich) were used as a positive control for mitochondrial depolarisation. Mitochondrial mass was measured by incubating cells with 100 nM MitoTracker Green (ThermoFisher) in serum-free media for 30 min at 37 °C. MSCs were then washed, detached and mass analysed. Mitochondrial superoxide generation was quantified by incubating MSCs with 5 µM MitoSOX Red in HBSS for 10 min at 37 °C, washing, detaching and measuring intracellular fluorescence.

Glucose uptake was measured by culturing cells in glucose-free basal media for 2 h before adding the fluorescently tagged glucose analogue 2-NBDG (2-(N-(7-Nitrobenz-2-oxa-1,3-diazol-4-yl) Amino-2-Deoxyglucose) for 60 min at 37 °C (ThermoFisher). MSCs were washed, detached and uptake of glucose measured by fluorescence incorporation.

Flow cytometry files were analysed using FlowJo software (version 10.5.3, FlowJo LLC, USA). In all experiments, a minimum of 5000 cells per sample were analysed to ensure statistical significance.

### Immunostaining

Topographies were washed twice with PBS, and cells fixed in a solution of 4% paraformaldehyde for 15 min at 37 °C. Fixed cells were permeabilised for 5 min at 4 °C in perm buffer (50 mM NaCl, 30 mM sucrose, 3 mM MgCl$_2$·6H$_2$O, 20 mM HEPES, 0.5% v/v Triton X-100 in PBS, pH 7.2) followed by blocking for 60 min at RT in blocking buffer (1% BSA (v/v) in PBS). Cells were stained with rhodamine tagged phalloidin (ThermoFisher) for 1 h in blocking buffer at RT. Nanotopographies were

washed for 6 × 5 min with wash buffer (PBS + 0.5% (v/v) Tween© 20 (Sigma-Aldrich)) and mounted with immunomount solution containing DAPI to stain nuclei (Vectorshield, Vector Laboratories). Cells were imaged using an Axiophot fluorescence microscope using ×10 and ×40 Neofluor objectives (all Zeiss), and captured with an Evolution QEi digital camera (Media Cybernetics, Rockville, USA) using QCapture software (QCapture Suite Plus, version 3.1.3.10, Teledyne QImaging, Surrey, BC, Canada). Actin expression was quantified using ImageJ software (version 1.52a) and normalised to nuclei count per field. 15 fields per topography were analysed.

To assess active caspase enzymes, cells were fixed, permeabilised and blocked as above and blocked with blocking buffer (1% milk protein in PBS with 0.1% (v/v) Tween 20). Cells were stained with primary antibodies against active caspase-3 (mouse anti-human clone 31A1067, Novus Biologicals) or active caspase-6 (Rabbit anti-human polyclonal, Novus Biologicals) overnight at 4 °C in blocking buffer. Cells were washed five times with PBS supplemented with 0.1% (v/v) Tween 20 (PBST). As normalization controls, CellTag700 stain (LI-COR) was diluted in blocking buffer (1:1000). To this, relevant secondary antibodies were diluted (LI-COR) and incubated for 1.5 h at room temperature with gentle agitation. Cells were washed five times with PBST and quantitative spectroscopic scanning and analysis carried out using the LI-COR Odyssey Sa. For analysis, internally normalised fluorescent intensities were normalised against flat controls to generate fold change fluorescent intensities.

### Super resolution microscopy

After immunostaining as above (monoclonal anti-Tomm20 (Abcam), phalloidin), samples were ready for super-resolution microscopy. Structured Illumination microscopy (SR-SIM) was performed using a Zeiss Elyra PS.1 super-resolution microscope (Carl Zeiss, Germany). A plan-Apochromat 63x/1.4 Oil lens was used, and Z-steps of 0.2 µm were acquired (total thickness between 5 and 7 µm) in five rotations using the ZEN Black Edition Imaging software.

The co-localisation of mitochondria with actin filaments was analysed in FIJI. Images were transformed to 8-bit and then inputted in the JACoP plugin for co-localisation[63]. Manders' coefficient and overlapping coefficient were used for the analysis. A threshold was set manually to measure the pixel intensity for each image and kept constant for the analysis of all the images.

### Seahorse mitochondrial flux measurements

MSCs were pretreated with metabolites for 14 days before being replated at $3 \times 10^4$ cells per well of a Seahorse XF 24-well plate overnight in standard culture medium. Cells were acclimatised for 1 h in phenol red free DMEM media containing 1 mM pyruvate, 2 mM glutamine and 10 mM glucose (all Agilent) before analysis. The mitochondrial stress test was performed according to the manufacturers protocol using 0.5 µM oligomycin, 1 µM FCCP and 0.5 mM rotenone/actinomycin A and data collected with a XFe24 analyser (all Agilent). Data were analysed and presented using Wave desktop software (Agilent).

### Cell culture secretome measurements

Extracellular lactate was measured from MSCs cultured on nanotopographies for 11 days in normal basal media. Cells were then washed twice with serum-free DMEM and cultured for the final 72 h in basal media where normal FCS was substituted with 10% (v/v) dialysed FCS (ThermoFisher). Cell culture supernatants were collected, aliquoted and frozen at −80 °C. Lactate levels in supernatants were quantified using the Lactate-Glo chemiluminescence assay (Promega), and luminescence measured using a Pherastar FS plate reader (BMG Labtech).

MSCs were primed for 72 h with 5 ng/ml IFNγ and 5 ng/ml TNFα and cell culture supernatants collected. Measurements of secreted IL-8 and TGFβ were evaluated by ELISA (Quantikine kits, R&D Systems)

according to manufacturer's instructions. PGE2 was measured using a Parameter ELISA kit (R&D Systems) according to manufacturer's instructions. L-Kynurenine in supernatant was measured using Erhlich's solution. Briefly, a standard curve of L-Kynurenine was generated using recombinant L-Kynurenine. 120ul of standard or sample was precipitated with 60 μl 30% Trichloroacetate solution and incubated at 50 °C for 30 min before centrifuging at $3000 \times g$ for 10 min. 75 μl of supernatant were added to duplicate wells of a 96-well plate. 75 μl of fresh Ehrlich's reagent (0.2 g p-Dimethylaminobenzaldehyde in 10 ml glacial acetic acid) was added per well and incubated for 15–30 min. Colour changes were read at 492 nm using a Multiscan GO plate reader (ThermoFisher).

To induce osteoblastic differentiation, cells were cultured for 21 days in osteoblastic media (complete DMEM media supplemented with 100 nM dexamethasone, 200 μM Ascorbate-2-phopshate and 10 mM β-glycerophosphate disodium salt hydrate) changing the media every 3 days. Cells were then fixed with 4% paraformaldehyde, washed with $dH_2O$ before covering cells with 2% Alizarin Red solution and incubating for 30 min in the dark. Cells were then washed 2–3 times with $dH_2O$ before imaging. To induce adipogenic differentiation, cells were cultured for 12 days in adipogenic media (complete DMEM media supplemented with 1 μM dexamethasone, 500 μM 3-isobutyl-1-methylxanthine, 1.72 μM insulin and 100 μM indomethacin) changing the media every 3 days. Cells were then fixed with 4% paraformaldehyde, washed with $dH_2O$, and covered with 60% isopropanol for 5 min. Isopropanol was then removed, and cells covered in working oil red o staining solution (300 mg oil red o in 100 ml 99% isopropanol, then diluted to a working stock of 3 parts oil red o solution to 2 parts $dH_2O$ and filtered to remove particulates) for 5–10 min. Cells were then washed twice with $dH_2O$ before imaging. In both cases, cells were imaged using a ×20 objective and an EVOS M5000 imaging system (Invitrogen).

## Gene expression

After culture, RNA was isolated from MSCs by lysing cells in 350 μl Trizol reagent (Life Technologies). 200 μl/ml of chloroform (Sigma) was added per 1 ml of Trizol, mixed, centrifuged and the aqueous phase removed. Total RNA was extracted using the RNeasy extraction kit (Qiagen) according to the manufacturer's protocol. Purified RNAs were quantified using a Nanodrop ND1000 spectrophotometer (Thermo Scientific) and cDNA synthesised with the Qiagen Quantitect reverse transcription kit, according to the manufacturer's protocol. PCR amplification of target genes was performed using Quantifast SYBR green qPCR kit (Qiagen) with specific primers (Eurofins). Primer sequences can be found in Supplementary Table 3. PCR was quantified using the $2^{-\Delta\Delta Ct}$ method and amplification performed using an Applied Biosystems 7500 Real Time PCR System and the fold upregulation of genes was compared to untreated controls.

## Immunosuppression assay

Peripheral blood mononuclear cells (PBMC)s were purified from buffy coats by layering onto Ficoll Plus (GE Healthcare) density gradients as per the manufacturer's instructions. PBMCs were labelled with 5 μM CellTrace CFSE (ThermoFisher) in PBS for 20 min at 37 °C. Labelled PBMCs were counted and resuspended in stimulation media (basal media containing 5 μg/ml Phytohemagglutinin (PHA)-P (Sigma-Aldrich) and 100 U/ml IL-2 (PeprotechEC)) to induce T lymphocyte proliferation. Stimulated CFSE-labelled cells were added to MSCs at a 4:1 ratio (4 PBMC to 1 MSC) unless otherwise stated. Cocultures were incubated for five days, before passing suspended cells through a 100 μm cell filter for analysis proliferation by CFSE dilution. Proliferation controls included in each assay were CFSE-labelled PBMCs in stimulation media alone (positive control), and CFSE-labelled PBMCs in basal media only (negative control). In order to compare independent assays, the proliferation index was calculated

using the proliferation analysis plugin in FlowJo software (version 10.5.3, FlowJo LLC, USA).

For inhibitor studies, Stro-1$^+$ cells were either pre-incubated for 1 h with 20 μM Jnk inhibitor (SP600125) or 10 μM ERK inhibitor (U0126) before washing four times with media and co-culturing with CFSE-labelled PBMCs as above. Alternatively, MSCs and CFSE-labelled PBMC co-cultures were setup in the presence of either 500 μM IDO1 inhibitor (1-MT), 20 μM PGE2 inhibitor (Indomethacin) or 1 mM nitric oxidase synthase inhibitor (L-NAME). Control wells of stimulated PBMCS alone with inhibitors were performed to check for direct effects on T cell proliferation.

## Metabolomic analysis

Whole-cell metabolomic analysis was performed on cell lysates isolated from MSC cultured on nanotopographies for 7 or 28 days. Nanotopographies were washed with ice-cold PBS, and cells lysed in extraction buffer (PBS/methanol/chloroform at 1:3:1 ratio) for 60 min at 4 °C with constant agitation. Lysates from duplicate nanotopographies were pooled as one sample, and extracts were transferred to cold eppendorfs and spun at $13,000 \times g$ at 4 °C for 5 min to remove debris. Cleared extracts were used for hydrophilic interaction liquid chromatography-mass spectrometry analysis (UltiMate 3000 RSLC (ThermoFisher), with a $150 \times 4.6$ mm ZIC-pHILIC column running at 300 μl/min$^{-1}$ and Orbitrap Exactive). Sample protein concentrations were measured by Nanodrop and used to standardise samples where required. A standardised pipeline, consisting of XCMS (peak picking), MzMatch (filtering and grouping) and IDEOM (further filtering, post-processing and identification) was used to process raw mass spectrometry data. Target metabolites identified were validated against a panel of unambiguous standards by mass and predicted retention time. Further putative identifications were generated by mass and predicted retention times. Means and standard errors of the mean were generated for every group of picked peaks and the resulting metabolomics data were uploaded to Ingenuity Pathway Analysis (IPA, Qiagen) software for pathway analysis. Heat maps of selected metabolites were generated using MetaboAnalyst software (version 4.0[64]).

## $^{13}C_6$-Glucose metabolomic tracing

MSC were seeded on nanotopographies at 1000 cells/cm$^2$ and allowed to grow for 11 days. Cells were washed and incubated in basal media comprising 50% normal glucose and 50% $^{13}C_6$-Glucose (Cambridge Isotopes Ltd) for a further 3 days. Extractions were performed as above. LC-MS was performed as previously described[69]. Briefly, the LC-MS platform consisted of an Accela 600 HPLC system combined with an Exactive (Orbitrap) mass spectrometer (ThermoFisher). Two complementary columns were used; the zwitterionic ZIC-pHILLIC column (150 mm × 4.6 mm; 3.5 μm, Merck) and the reversed phase ACE C18-AR column (150 mm × 4.6 mm; 3.5 μm Hichrom) and in both cases sample volume was 10 μl at a flow rate of 0.3 ml/min. Eluted samples were then analysed by mass spectrometry.

LCMS data of $^{13}C$-labelled extracts were processed to generate a combined PeakML file as described previously[65]. Further analysis using mzMatch-ISO in R[66] generated a PDF file containing chromatograms used to check peak-shape and retention time. A tab-delineated file detailing peak height for each isopotologue was also generated to calculate percentage labelling.

## Next-generation sequencing (RNAseq)

RNA was isolated from MSCs using the RNeasy extraction kit (Qiagen) following the manufacturer's instructions. RNA concentrations were quantified using a Nanodrop ND1000 spectrophotometer (Thermo-Fisher) and samples submitted to the Glasgow Polyomics facility (University of Glasgow). 136 ng of RNA per sample was processed (Truseq$^©$ kit, Illumina) and a template mRNA library generated (poly-A selection, first and second strand cDNA synthesis, cleanup,

adenylation of 3′ ends, ligation and cleaning of adaptors). Amplification (98 °C for 30 s, then 15 cycles of; 98 °C for 10 s, 60 °C for 30 s and 72 °C for 30 s followed by 72 °C for 5 min) was performed and PCR products purified with AMPure XP beads (Beckman Coulter). 3.2 pM libraries were sequenced (Nextseq 500 analsyer, Illumina) and results processed using the BaseSpace© next-generation sequencing platform (Illumina), with differential expression data being analysed with IPA software (Qiagen).

## Western blotting

MSCs were cultured in 25 cm² flasks in the presence or absence of metabolites or Y-27632 inhibitor. Cells were washed once with cold PBS, then lysed in 200 µl RIPA buffer (150 mM NaCl, 1% (v/v) NP-40, 0.1% (v/v) Sodium dodecyl sulfate in 50 mM Tris HCl pH 8) supplemented with protease and phosphatase inhibitors (Pierce Inhibitor Mini Tablets, ThermoFisher). Lysate protein concentration was measured (Pierce BCA Protein Assay Kit, ThermoFisher) and 20 µg loaded per lane of a Bolt 4–12% Bis-Tris Gel using a XCell SureLock gel tank (both ThermoFisher). Protein was transferred onto Immobilon-FL PVDF membrane (Merck) and blocked for 1 h at RT with 5% (v/v) nonfat dry milk powder in Tris Buffered Saline (TBS). Membranes were probed overnight at 4 °C with mouse anti-human phospho-myosin light chain 2 (Ser19, clone #3675, Cell Signalling Technology) and mouse anti-human β-tubulin (clone TUB 2.1, Sigma-Aldrich) in TBS with 5% (v/v) BSA. Membranes were washed with TBST (TBS + 0.5% (v/v) Tween-20) for 6 × 5 min, and probed with IRDye 800CW goat anti-mouse IgG secondary antibody (Li-Cor) in TBS containing 5% (v/v) BSA for 1 h at RT. Membranes were washed for 6 × 5 min and imaged using an Odyssey Sa infrared imaging system with Image Studio 4.0 software (Li-Cor). Bands were quantified by densitometry using ImageJ software[63].

## Single-cell nanoindentation

Mechanical characterisation of the cells was achieved through nanoindentation. Measurements were performed using a nanoindenter (Chiaro, Optics11, Amsterdam, NL) placed on top of an inverted phase contrast microscope (EVOS XL Core, Thermofisher). Nanoindentation was performed 7 days post cell seeding on the respective substrates and after an overnight serum starvation performed before the measurements to promote cell cycle synchronisation. Optical and geometrical calibrations were performed according to the manufacturer's instructions. A spherical tip of a 9 µm radius, attached to a cantilever with a spring constant 0.025 N/m was used for performing measurements. Indentation curves were obtained at a speed 2 µm/s over a vertical range of 10 µm. For each cell, 9 curves were collected over a 3 × 3 map of curves with a 500 nm pitch, and the average of the values used as the representative value for the single cell. The curves were analysed following a protocol previously described[67]. In brief, the curves were filtered, the contact point was identified using a direct thresholding method, and finally fitted with a standard Hertz model keeping the maximum indentation under 10% of the radius. All the data analysis was performed with a set of open Python libraries developed in the lab and available open source.

## Statistics

Unless stated, two-tailed unpaired T tests (Mann–Whitney) were performed where a direct comparison was assessed between two population groups using Prism software unless otherwise stated (GraphPad). A sample population of between 3 and 7 replicates was always used. Results are quoted as mean ± standard error of the mean or standard deviation, where appropriate. The probability values are quoted to an accuracy of 95%, 99% and 99.9% (*$P < 0.05$, **$P < 0.01$ and *$P \leq 0.001$, respectively). Supplementary Table S4 indicates sample numbers and replicates in each figure presented.

## Reporting summary

Further information on research design is available in the Nature Portfolio Reporting Summary linked to this article.

## Data availability

All data supporting the findings in this study are available within the article and its Supplementary Information files, can be obtained from the corresponding author or can be accessed at: http://researchdata.gla.ac.uk/973/. Source data are provided with this paper.

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

## Acknowledgements

We thank Carol-Anne Smith for technical assistance. This work was supported by BBSRC funded grants BB/N018419/1, BB/K011235/1 and BB/L021072/1.

## Author contributions

E.A.R., L.-A.T., M.S.-S., M.V., J.M. and M.J.D. conceived and designed the analysis. E.A.R., L.-A.T., H.D., M.P.T., V.J., Y.H., M.A.G.O., J.W., Y.X. and J.B., performed the experimental work. A.S., P.R. and N.G. manufactured and provided polycarbonate nanotopographies. K.V.B. and G.B. performed metabolomic profiling and preliminary analysis of untargeted and $^{13}$C labelled cell extracts. J.A.W. and R.O.C.O. provided Stro-1$^+$ cells. E.A.R. and M.J.D. wrote the manuscript. L.-A.T., G.B., M.S.-S., J.M., N.G., M.V., M.A.G.O., V.J., M.P.T. and R.O.C.O. revised the manuscript and were involved in the discussion of the work. N.G., R.O.C.O. and M.J.D. secured funding.

## Competing interests

The authors declare no competing interests.
