## [Peer Review File · Nature Communications]

Editorial Note: This manuscript has been previously reviewed at another journal that is not operating a transparent peer review scheme. This document only contains reviewer comments and rebuttal letters for versions considered at *Nature Communications*

REVIEWER COMMENTS

Reviewer #2 (Remarks to the Author):

To date, the following points add to the major open questions that are still hindering substantial progress in the MSC field:

- a. Which are the relevant mechanisms that drive MSC immunomodulation in vivo?**
- b. How to develop robust and reproducible manufacture technologies that maintain MSC immunomodulation capacities?**
- c. How to design assays that reliably predict MSC immunomodulation in vivo?**

In their revised manuscript entitled „Nanotopography reveals metabolites that maintain the immunomodulatory phenotype of mesenchymal stromal cells“ the authors report, exclusively based on in vitro studies, that the potential of human Mesenchymal Stromal Cells (MSCs) to suppress immune cell proliferation can be modulated by engineering of the culture matrix leading to changes in intracellular tension, and thus, in oxidative glycolysis. Hereby, they identified metabolites that, added to the culture medium, can maintain the potential of MSC cultures to suppress immune cell proliferation in vitro.

The revised manuscript aims to address some interesting mechanistic aspects of MSC immunomodulation placing their metabolic state in the center of the concept which is a relatively novel approach. Also, the quality of the metabolomic analyses and their thoughtful applications can be acknowledged.

Compared to the initially submitted manuscript the authors performed some of the reviewer´s suggested experiments, and they included and discussed these data in the revised manuscript:

- a. Conduct experiments to compare the effects of SQ MSC to addition of lactate and low oxygen partial pressure (e.g. 2%) and discuss the above mentioned papers in the Discussion section.**
- b. Analyze the distribution of electrical charges on SQ vs flat surfaces.**
- c. Add at least one in vitro assay to assess potential MSCs immunomodulation capacity (killing assay)**

I have the following comments referring to the major claims mentioned in the revised manuscript

Claim #1: “Nanotopography can maintain MSC immunomodulatory capacity via decreased intracellular tension”

The here presented data support the authors´ claims that the physiological differences observed are likely through modulation of cell behaviour via their contact with the different surfaces and not due to an intrinsic difference in the properties of the materials/surfaces being used. However, their claim that “this is the first time a material alone has been shown to enhance the immunomodulatory capability of MSCs” may not be upheld (reviewed in Kouroupis D and Correa D, Front. Bioeng. Biotechnol., 2021

<https://www.frontiersin.org/articles/10.3389/fbioe.2021.621748/full>
| <https://doi.org/10.3389/fbioe.2021.621748>). One might credit the authors that this review might have been published after/during preparation of the revised manuscript; still it is suggested to discuss this reference and to consider re-phrasing. It appears that the current manuscript is an extension of the authors' previously developed concept on the axis culture surface/matrix – tension modulation – MSC function. Thus, particularly considering the mild impact on MSC immunomodulation in vitro (see below), it might be beneficial to focus on the detailed metabolic findings that are presented here, rather than over-stressing the translational impact for the production of MSC for immunotherapies, where, in the reviewers' point of view, strong evidence is still lacking as long as supportive in vivo data, or striking in vitro data, cannot be added.

The authors observed that growing MSCs on SQ decreased their intracellular tension. They also observed that these MSCs had an increased (though only mildly) potential to suppress PBMNC proliferation. However, a clear mechanistic connection between these observations was still not delivered. Such, the proposed causal chain is not supported by the currently presented data.

Claim #2: "Untargeted metabolomics points to respiration as a central mechanism"

The concept of metabolic activity being involved in MSCs' immunomodulation potential is intriguing and, thus, warrants detailed exploration. The authors have put quite some effort into this performing additional experiments as previously suggested by the reviewer. Particularly, their conclusion that MSC growth on SQ topography reduces their mitochondrial activity without affecting their organelle physiology was carved out nicely.

Yet, the following concerns prevail:

a. SQ-MSCs produced more lactate, but adding lactate to MSCs grown on flat surface (as per reviewer's suggestion) did not increase the potential of these MSCs to suppress PBMNC proliferation in vitro (Fig. S8b). Adding lactate actually increased MSC membrane stiffness (Fig. S8a), which is quite the opposite one would have expected if, as per the authors suggested overall concept, reduced membrane stiffness is a mechanistic key factor for promoting MSCs' immunomodulatory functions.

Moreover, even if one might be inclined to follow the authors' claim of seeing a "trend" towards elevated apoptosis for MSCs cultured on the SQ nanotopography with T-cell co-culture (Fig. S3b and c)" as a surrogate for improved MSC immunomodulation, the same "trend" might pertain to NSQ group that, per the data presented by the authors, featured increased membrane stiffness (Fig. 1g) and lower potential to suppress PBMNC proliferation in vitro.

The authors are that their findings may conflict with previously published data (Selleri S et al. *Oncotarget*. 2016; 7:30193-30210.) that reported on MSC lactate production as a mechanism of their immunomodulation capacity. Yet, instead of critically reflect their own data, the authors suggest (without clear evidence) the different MSC sources (BM vs UC) as likely reason for the differences.

b. As suggested by the reviewer, the authors conducted additional experiments under hypoxic conditions (Fig. S5). They observed reduced membrane tension also under hypoxia in the SQ group compared to the flat surface group; however, the relevant non-hypoxic control is missing here. Moreover, no difference regarding the capacity to reduce PBMNC proliferation was observed which would have been a relevant point, as to the authors' logic regarding the relevance of this in vitro assay, to support the authors' claim that membrane

stiffness is a relevant mechanistic key player in MSC immunomodulation.

Claim #3: Bioactive metabolites can enhance immunomodulatory capacity and growth; Large scale growth of immunomodulatory MSCs

The authors report that when MSCs were treated with metabolites and subsequently co-cultured with pre-activated PBMCs there was no difference in susceptibility to apoptosis in any of the treated cells compared to controls (Fig. S10b). They conclude that this suggests "metabolite priming MSCs to promote immunomodulation activities retains their ability to respond to activated T cells and induce apoptosis in agreement with studies showing that this MSC physiological effect is correlated to potential therapeutic efficacy". However, the shown differences for PBMC proliferation suppression were only mild, and together with the lack of impact on MSC killability by activated immune cells as a second mechanism, the concept of controlling MSC immunomodulation by their biophysical properties (stiffness, culture conditions) is not strongly supported.

In addition, it appears contradictory that MSC would proliferate well, or at least sufficiently, for clinical grade production, when pro-apoptotic enzymes such as Caspase 3 and Caspase 6 were up-regulated.

In summary, it is appreciated that the authors added some of the suggested experiments that, however, came out with mixed results; such, the following major concerns prevail:

a. Many mechanistic unknowns persist

b. The key point, i.e. relevant impact on MSCs' immunomodulation capacity, has still not been sufficiently addressed:

- MSC-PBMC co-culture assay performed showing only mild differences
- No difference in IDO production; other known mediators (e.g. TSG-6) were not investigated
- Induction of Caspase 3 and Caspase 6 was shown, but no effects in the killing assay were observed
- Contradictory results for lactate exposure (increase of membrane stiffness, but not the expected decrease; no effect on PNBMC proliferation)
- Above mentioned issues with results from hypoxia experiments
- No Treg induction (the authors' claim that treated MSCs increased Tregs is not comprehensible, as the presented data does not support it (Fig. S11b; bars are very similar; no statistically significant difference reported)
- Translational relevance still missing (lacking in vivo data as pointed out in the first round review)

Minor points:

The summary of donors and replicates (Table S4) is helpful, but it is not always clear how it corresponds to the text (e.g. p.7; l. 204: "we use both an enriched skeletal (Stro-1+) MSC population and an unselected (total) commercial bone marrow-derived skeletal MSC population"; were these from 7 donors, or 3 donors, or, as it sounds, only one, or maximally 2 donors?).

In the point-to-point response a different version of Fig. S8 (including a "Metabolites" group) is shown compared to the manuscript.,

Suggestions:

a. Create a graphical abstract illustrating the suggested concept and interactions of key mechanistic players starting from SQ impact on MSC membrane stiffness, highlighting the

mechanistic links to MSC immunomodulation factors.

This would not only help readers to follow better the authors' concept and to contextualize the data supporting their concept, but also show the authors where they could add evidence to support their claims and, thus, to tap the (possible) full potential of their findings.

b. Clearly identify the molecular link(s) how changes in cell tension impact oxidative glycolysis and, if proven, immunomodulation.

c. Validate the MSC immunomodulation potential at least in one in vivo model as this is currently still based solely on a single in vitro assay, i.e. suppression of PBMNC proliferation.

Reviewer #3 (Remarks to the Author):

The authors addressed my concerns

Reviewer #5 (Remarks to the Author):

This manuscript has already undergone one round of peer review and the authors have made an extensive revision, including a robust number of new experiments and data, based on the comments of four reviewers. Three out of four reviewers are supportive to the study while the fourth considered that "the study is severely flawed by a number of issues", which I respectfully disagree, especially based on the revised version of the manuscript. According to the authors, "the central aim of this study was to demonstrate that nanotopography can be used to identify pathways that can be exploited to maintain the immunomodulatory capabilities of MSCs in culture for a prolonged period of time", and they clearly succeeded in proving it. Considering the revision process, the inclusion of Figure S2, as suggested by Reviewer 1, that showed that the total surface energy of the topographies is not different between the different patterns, proving that "physiological differences observed are through modulation of cell behaviour via their contact with the different surfaces and not due to an intrinsic difference in the properties of the materials/surfaces being used" is interesting and of relevance to the goal of the study. The authors explored in very good way a fair point raised by Reviewer 2 regarding the originality of the study. While a PubMed search of "nanotopography and MSC" produced 34 hits, the search of 'nanotopography and MSC and immunomodulatory', the claim authors make in this study, yields 0 hits, demonstrating the novelty of this work. The hypothesis elaborated by the authors in response to Reviewer 3 to explain the relationship between the level of intracellular tension and cell differentiation and proliferation are plausible and smart. I agree with Reviewer 3 on the relevance of elucidating "the molecular link(s) how changes in cell tension impact oxidative glycolysis.", mainly the role of actin. However, as the authors mentioned, this subject would generate data for a whole new article. The inclusion of caspase, apoptosis and Treg assays as well as the T cell proliferation satisfied points made by Reviewers 3 and 4. In summary, my only concern on this manuscript is the lack of in vivo data as pointed out by Reviewers 2 and 3 since the inflammatory environment is far more complex than the controlled in vitro one. However, as mentioned in the rebuttal letter "The editor at Nature Biomedical Engineering felt in vivo data was important for them but spoke to the editors at Nature Communications who agreed that the work had sufficient novelty in the in vitro data to be of interest to them.". Thus, I considered that the approaches the authors used are appropriate and sufficient to support their conclusions with robust and consistent data and that the points raised by the Reviewers have been

effectively addressed. In this context, I believe that this manuscript exhibit the high quality and novelty to justify its publication in Nature Communications.

Response to the Reviewer.

We thank the reviewer for their suggestions throughout the review process and for clarifying some of our questions at the last review stage. We have performed a range of new experiments looking at soluble factor release (showing PGE2 and IL-8 are increased), inhibitor studies (showing that IDO-1 is important), TSG-6 analysis (showing MSCs maintained their ability to induce TSG-6 gene expression when exposed to TNF), multipotency of the MSCs post metabolite treatment (showing they retain multipotency), mitochondrial-cytoskeleton association studies (showing mitochondria become less associated with cytoskeleton when they retain immunomodulatory phenotype) – all to help show the cytoskeleton-respiration-immunomodulation axis. We provide a point-by-point response to their questions below. Again, we thank the reviewer for their time and consideration and we hope that this new version, including significant new data, will mean that they find our work to be publishable in Nature Communications.

To date, the following points add to the major open questions that are still hindering substantial progress in the MSC field:

- a. Which are the relevant mechanisms that drive MSC immunomodulation in vivo?
- b. b. How to develop robust and reproducible manufacture technologies that maintain MSC immunomodulation capacities?
- c. How to design assays that reliably predict MSC immunomodulation in vivo?

In their revised manuscript entitled „Nanotopography reveals metabolites that maintain the immunomodulatory phenotype of mesenchymal stromal cells“ the authors report, exclusively based on in vitro studies, that the potential of human Mesenchymal Stromal Cells (MSCs) to suppress immune cell proliferation can be modulated by engineering of the culture matrix leading to changes in intracellular tension, and thus, in oxidative glycolysis. Hereby, they identified metabolites that, added to the culture medium, can maintain the potential of MSC cultures to suppress immune cell proliferation in vitro.

The revised manuscript aims to address some interesting mechanistic aspects of MSC immunomodulation placing their metabolic state in the center of the concept which is a relatively novel approach. Also, the quality of the metabolomic analyses and their thoughtful applications can be acknowledged.

We thank the reviewer for acknowledging the novelty and importance of our research.

Compared to the initially submitted manuscript the authors performed some of the reviewer’s suggested experiments, and they included and discussed these data in the revised manuscript:

- a. Conduct experiments to compare the effects of SQ MSC to addition of lactate and low oxygen partial pressure (e.g. 2%) and discuss the above mentioned papers in the Discussion section.
- b. Analyze the distribution of electrical charges on SQ vs flat surfaces.
- c. Add at least one in vitro assay to assess potential MSCs immunomodulation capacity (killing assay)

I have the following comments referring to the major claims mentioned in the revised manuscript

Claim #1: “Nanotopography can maintain MSC immunomodulatory capacity via decreased intracellular tension”

The here presented data support the authors’ claims that the physiological differences observed are likely through modulation of cell behaviour via their contact with the different surfaces and not due to an intrinsic difference in the properties of the materials/surfaces being used. However, their claim that “this is the first time a material alone has been shown to enhance the immunomodulatory capability of MSCs” may not be upheld (reviewed in Kouroupis D and Correa D, Front. Bioeng. Biotechnol., 2021 <https://www.frontiersin.org/articles/10.3389/fbioe.2021.621748/full> | <https://doi.org/10.3389/fbioe.2021.621748>). One might credit the authors that this review might have been published after/during preparation of the revised manuscript; still it is suggested to discuss this reference and to consider re-phrasing. It appears that the current manuscript is an extension of the authors’ previously developed concept on the axis culture surface/matrix – tension modulation – MSC function. Thus, particularly considering the mild impact on MSC immunomodulation in vitro (see below), it might be beneficial to focus on the detailed metabolic findings that are presented here, rather than over-stressing the translational impact for the production of MSC for immunotherapies, where, in the reviewers’ point of view, strong evidence is still lacking as long as supportive in vivo data, or striking in vitro data, cannot be added.

The reviewer directs us to a nice review article focussing on spheroids and soluble factors. However, the review does not really focus on the cell-material interface or immune modulation – the areas our article targets. Thus, we feel our claim that our result is a materials-driven first stands. However, we are happy to remove the statement in light of the reviewer’s concern.

The authors observed that growing MSCs on SQ decreased their intracellular tension. They also observed that these MSCs had an increased (though only mildly) potential to suppress PBMNC proliferation.

As we have discussed in previous rebuttals, we disagree that suppression is mild. The proliferation index analysis makes it look mild, but a significant change represents a large shift from a non-responsive MSC population to a more naïve population with many immunomodulatory cells left within the population, as shown in fig. 6 c&d.

However, a clear mechanistic connection between these observations was still not delivered. Such, the proposed causal chain is not supported by the currently presented data.

The reviewer suggested that inhibition experiments could help to highlight biological mechanism. We performed a range of inhibitions and have shown IDO1 to be a driving mechanism. Thus, we have new main text and a new supplementary figure, as shown below.

Functional, immuno-modulatory MSCs have a number of strategies to dampen an immune response including the release of paracrine soluble factors (e.g. transforming growth factor (TGF β), hepatocyte growth factor (HGF), prostaglandin E2 (PGE2), prostaglandin-endoperoxide synthase (Cox-2), nitric oxide (NO), IL-6 etc) as well as the release of enzymes, extracellular vesicles and receptor-mediated cell-cell contacts⁵⁰. We, therefore, examined the secretome of metabolite-treated MSCs after priming with the pro-inflammatory mediators tumour necrosis factor (TNF α) and interferon gamma (IFN γ)⁵¹. Firstly, we observed that secretion of IDO1 is the main immunomodulator of T cell suppression in our co-culture (Fig S14). Blocking with the established IDO1 inhibitor 1-MT reduced the ability of MSCs to suppress proliferation. Blocking other established immunomodulatory pathways (Cox-2, PGE-2, NO)⁵¹ had little impact on T cell suppressive properties (Fig. S14a). We also inhibited biochemical pathways that have been implicated in MSC growth, extracellular signal-related kinase (ERK) and c-Jun N-terminal kinase (JNK)^{3,19}; neither were observed to have an effect on T-cell proliferation index in co-culture (Fig. S14a). Secondly, we performed the T cell suppression assay with MSCs treated with both the mixed metabolites in the presence of 1-MT IDO1 inhibition. Here, the metabolite-enhanced MSC immuno-modulatory effect became negligible, again showing that the metabolite-driven T cell suppression is mainly mediated through IDO1 (Fig. S14b).

Fig. S14. Effects of pathway inhibition on Stro-1⁺ MSC immunosuppression. (a) Stro-1⁺ MSCs were treated with inhibitors of IDO1 activity (1-MT), Cox2 (Indomethacin), Nitric Oxide synthase (L-NAME), pan JNK inhibitor (SP600125) or ERK 1/2 inhibition (U0126). Effects on the cells ability to suppress T cell proliferation were then assessed through CFSE dilution by flow cytometry. (b) Stro-1⁺ MSCs were treated with niacinamide + metabolites, lactate or Y27632 for 7 days before co-culturing with CFSE labelled PBMCs in the presence of 1-MT for 5 days. T cell proliferation was assessed through CFSE dilution using flow cytometry, and effects on immunosuppression quantified. Graphs show mean \pm SEM (n=3 co-cultures per donor) of two donors.

Claim #2: “Untargeted metabolomics points to respiration as a central mechanism”

The concept of metabolic activity being involved in MSCs` immunomodulation potential is intriguing and, thus, warrants detailed exploration. The authors have put quite some effort into this performing additional experiments as previously suggested by the reviewer. Particularly, their conclusion that MSC growth on SQ topography reduces their mitochondrial activity without affecting their organelle physiology was carved out nicely.

We thank the reviewer for these positive comments and we have done a little more by looking at mitochondrial-actin association with some new main text and a new supplementary figure, as shown below.

Finally, as we are relating mitochondrial activity to intracellular tension we used image analysis of mitochondria and actin and employed Manders and overlapping coefficients to look at mitochondrial association with actin; both calculate pixel co-localisation³⁵. While MSCs on control and NSQ surfaces had similar coefficients, the co-localisation of mitochondria and actin was reduced in MSCs grown on the SQ topography and with Y27632 treatment, with the effect of the SQ nanotopography being more subtle than Y27632 treatment (Fig. S8).

Fig. S8. Effects of nanotopography on mitochondrial distribution in MSCs. (a) Cells were grown on topographies for 7 days before fixation and immunofluorescent staining for total mitochondria (anti-Tomm20 antibody; purple) and phalloidin-FITC to label actin. Super resolution microscopy was performed and the co-localisation of mitochondria with actin filaments was evaluated using image analysis (b and c). Graphs show the mean and S.E.M of where each point represents an independent field of view analysed. Statistical comparisons by ANOVA with Dunnett's test of multiple comparisons.

Yet, the following concerns prevail:

a. SQ-MSCs produced more lactate, but adding lactate to MSCs grown on flat surface (as per reviewer's suggestion) did not increase the potential of these MSCs to suppress PBMNC proliferation in vitro (Fig. S8b). Adding lactate actually increased MSC membrane stiffness (Fig. S8a), which is quite the opposite one would have expected if, as per the authors suggested overall concept, reduced membrane stiffness is a mechanistic key factor for promoting MSCs' immunomodulatory functions.

We feel that the correct trend is apparent and adding lactate caused a stiffer membrane and reduced MSC capacity for immunomodulation and we feel that this is in line with our hypothesis (that SQ reduced membrane tension and increases immunomodulatory phenotype, as in Fig. 1g). Why it increases tension we don't know and we believe that finding out goes beyond the scope of the paper.

Moreover, even if one might be inclined to follow the authors' claim of seeing a "trend" towards elevated apoptosis for MSCs cultured on the SQ nanotopography with T-cell co-culture (Fig. S3b and c)" as a surrogate for improved MSC immunomodulation, the same "trend" might pertain to NSQ group that, per the data presented by the authors, featured increased membrane stiffness (Fig. 1g) and lower potential to suppress PBMNC proliferation in vitro.

We agree with the reviewer that we dislike talking in 'trends'. That said, we think that the correct trend is apparent – the NSQ MSCs have higher stiffness and lower apoptosis (compared to SQ MSCs) while the SQ MSCs have lower stiffness and higher apoptosis.

The authors are that their findings may conflict with previously published data (Selleri S et al. *Oncotarget*. 2016; 7:30193-30210.) that reported on MSC lactate production as a mechanism of their immunomodulation capacity. Yet, instead of critically reflect their own data, the authors suggest (without clear evidence) the different MSC sources (BM vs UC) as likely reason for the differences.

*We feel that we have taken the reviewer's comments very seriously and they have helped us to reflect on many aspects of our work. The papers are hard to compare as aged (as we use) and young (as they use) MSCs are quite different cells with quite different growth profiles. Also, we look for different things and while we look at direct immunomodulation, they look at secreted markers. Other groups have reported differences in the immunomodulatory capacities of the two sources of cells (BM tend to use more IDO1 whilst UC tended to utilise PGE2; Song et al, *W J Stem Cells* 2020); thus although both sources of MSCs can immunosuppress, they utilise different mechanism. This, and other reports demonstrate subtle differences in the physiology of MSCs isolated from different tissues. We do not discount secretion of lactate as a mechanism of their immunosuppression, although this is predominantly reported to affect macrophage polarisation. This effect on macrophages in this context is ongoing work in our laboratory, and we feel is outside the scope of this current manuscript. We have now extended our data to include a study of secreted immunomodulatory markers TGF β , IL-8, PGE-2 and IDO-1 (via kynurenic acid production), showing that expression of IL-8 and PGE-2 were increased with MSCs cultured with our selected metabolites. We also showed that metabolite-treated MSCs maintained their ability to induce TSG-6 when exposed to TNF- α in a new supplementary figure. This new data further supports our hypothesis.*

Fig. S15. Measuring immunomodulatory factors in metabolite treated Stro-1⁺ MSC secretome. Stro-1⁺ MSCs were treated with metabolites, lactate or ROCK inhibitor Y27632 for 7 days then ‘primed’ for 72 hours with 5 ng/ml IFN γ and 5 ng/ml TNF α . Secretion of IL-6, IL-8 and PGE2 was measured by ELISA (a-c). IDO-1 activity was measured using Ehrlich’s solution to measure L-kynurine, a breakdown product of tryptophan (d). (e) Induction of TSG-6 gene expression after IFN γ and TNF α priming by qPCR (n=2 donors). (f) TSG-6 gene expression measured after IFN γ and TNF α priming in metabolite treated Stro-1⁺ MSCs compared to untreated cells. Graphs in a-d and f show mean \pm SEM of four independent donors. Comparisons by ANOVA with Dunnett’s test of multiple comparisons).

b. As suggested by the reviewer, the authors conducted additional experiments under hypoxic conditions (Fig. S5). They observed reduced membrane tension also under hypoxia in the SQ group compared to the flat surface group; however, the relevant non-hypoxic control is missing here. Moreover, no difference regarding the capacity to reduce PBMNC proliferation was observed which would have been a relevant point, as to the authors' logic regarding the relevance of this in vitro assay, to support the authors' claim that membrane stiffness is a relevant mechanistic key player in MSC immunomodulation.

Our data clearly shows that a hypoxic environment results in an increased immunosuppressive potential of the cells (as is sensible and known) and in these growth conditions, the influence of the topographies do not appear to further increase this physiological function. Therefore, even if membrane tension changes, no further change in immunosuppression is observed; i.e. maximal immunosuppressive potential has been reached in the low oxygen environment. However, the crucial point of this study is that the topographies, by themselves, in a normoxic, standard tissue culture environment, can promote MSC immunomodulatory capacity, and this effect is new to the field. We note that the normoxic and hypoxic experiments were run at the same time and so the data controls itself between Fig. 1g and Fig. S5a; it is noteworthy that the flat values for stiffness are essentially the same from hypoxia to normoxia.

Claim #3: Bioactive metabolites can enhance immunomodulatory capacity and growth; Large scale growth of immunomodulatory MSCs

The authors report that when MSCs were treated with metabolites and subsequently co-cultured with pre-activated PBMCs there was no difference in susceptibility to apoptosis in any of the treated cells compared to controls (Fig. S10b). They conclude that this suggests "metabolite priming MSCs to promote immunomodulation activities retains their ability to respond to activated T cells and induce apoptosis in agreement with studies showing that this MSC physiological effect is correlated to potential therapeutic efficacy". However, the shown differences for PBMNC proliferation suppression were only mild, and together with the lack of impact on MSC killability by activated immune cells as a second mechanism, the concept of controlling MSC immunomodulation by their biophysical properties (stiffness, culture conditions) is not strongly supported.

In addition, it appears contradictory that MSC would proliferate well, or at least sufficiently, for clinical grade production, when pro-apoptotic enzymes such as Caspase 3 and Caspase 6 were up-regulated.

We accept that the changes in killability in vitro were mild compared to in vivo observations. However, levels obtained in these experiments were comparable to those reported by Galleu and colleagues who defined this correlation between in vitro apoptosis induction and therapeutic efficacy (Galleu et al., Sci Trans Med 2017). It is intriguing that caspase levels were raised and this does suggest the cells may be primed to apoptosis in the right conditions; we agree that while the caspase data is strong, the apoptosis data is only gently confirmatory to our hypothesis. We also accept that our TReg data simply shows that our primed MSCs do not lose the ability to promote TReg differentiation and we have modified the text in the manuscript as the reviewer requests later in their review. However, this does not disprove our

hypothesis and demonstrates that our treated MSCs retain this important immunomodulatory functionality. In the last revision, we had ran three distinct functional assays that all help to support our hypothesis. In this new revision we add in also SFig 14 and 15 on inflammatory factors and IDO mechanism. Again, these support our hypothesis.

We also show that the MSCs grow and are viable in vitro despite having increased caspase levels – indeed we grow many more MSCs with the metabolite treatment compared to without. We think this must be the expected outcome for naïve MSCs – that they grow in culture with good viability but are primed to apoptose in the right condition – our paper now contributes to this hypothesis with the help of the reviewer.

In summary, it is appreciated that the authors added some of the suggested experiments that, however, came out with mixed results; such, the following major concerns prevail:

We have performed a lot of new analysis from the original paper in both revision 1 and 2. We are not sure we would say the data is mixed as it all points one way. We would agree that sometimes the analysis is not sensitive enough to pick out the differences between naïve MSCs cultured on SQ or with metabolites (although the trend is always there) and sometimes it is sensitive enough. Despite a lot of data, the results never go the wrong way. It is worth noting that this data has been generated with cells from many donors now and so these trends and significances, that are always confirmatory, are there despite the donor-to-donor variance. We believe our data to be robust and honest.

a. Many mechanistic unknowns persist

We have performed a number of inhibition studies that highlight IDO-1 as a key player.

b. The key point, i.e. relevant impact on MSCs' immunomodulation capacity, has still not been sufficiently addressed:

- MSC-PBMNC co-culture assay performed showing only mild differences
As we have mentioned, we disagree with this point and we have now, as described above, built a lot more data around this central finding.
- No difference in IDO production; other known mediators (e.g. TSG-6) were not investigated
We now show that while IDO production per se was not changed (beyond a trend), we show through IDO-1 inhibition that it is a key mediator of the immunomodulatory effect. TSG-6 has been investigated now but simply showed that MSCs maintained their ability to induce TSG-6 gene expression when exposed to TNF- α . We have attempted to directly measure secreted TSG-6 after 24 and 72 hours of TNF α priming. Unfortunately, using two separate commercial ELISA kits and MSCs from four individual donors we have been unable to detect any secreted TSG-6 from our cells. We now provide new PGE2 and IL-8 data, and also show a role for IDO-1 that significantly support our hypothesis.
- Induction of Caspase 3 and Caspase 6 was shown, but no effects in the killing assay were observed

We propose that the cells are more killable only in the right situation. Our data is also in line with that of Galleu et al., Sci Trans Med 2017, who originally described increased naïve MSC killability.

- Contradictory results for lactate exposure (increase of membrane stiffness, but not the expected decrease; no effect on PNBMC proliferation)
We think the data matches up to the hypothesis as we describe above.
- Above mentioned issues with results from hypoxia experiments.
Our focus is on normoxia as we discuss above. We note that the normoxic and hypoxic experiments were ran at the same time and so control each other.
- No Treg induction (the authors` claim that treated MSCs increased Tregs is not comprehensible, as the presented data does not support it (Fig. S11b; bars are very similar; no statistically significant difference reported)
We have toned down our language here as we simply wish to claim that the MSCs can still support Treg differentiation in all conditions.
“To demonstrate the direct functional activity of treated MSCs on other immune cells, the ability to promote regulatory T cell (TReg) programming during co-culture with T cells was also evaluated; this is a key immunomodulatory function of naïve MSCs (Fig. S16). Metabolite-treated cells were able to support TReg differentiation, as did MSCs in control conditions. These data suggest that treatment with metabolites will not impair their therapeutic potential during expansion in culture”.
- Translational relevance still missing (lacking in vivo data as pointed out in the first round review)
We have now performed a lot of new experiments that either show that the topography/metabolite treated MSCs enhance immunomodulation (proliferation index, cytokines, IDO-1 inhibition, caspase expression) or don't negatively affect an immune function (Treg differentiation, apoptosis, TSG-6). We also have editorial guidance as we noted previously that Nature Communications are not necessarily looking for in vivo data while Nature Biomedical Engineering were.

Minor points:

The summary of donors and replicates (Table S4) is helpful, but it is not always clear how it corresponds to the text (e.g. p.7; l. 204: “we use both an enriched skeletal (Stro-1+) MSC population and an unselected (total) commercial bone marrow-derived skeletal MSC population”; were these from 7 donors, or 3 donors, or, as it sounds, only one, or maximally 2 donors?).

We thank the reviewer for pointing this out and we have revised table S4 to reflect this. Essentially, 3 donor lines of the Stro-1 cells and 3 donor lines of the commercial whole adherent cells were used.

In the point-to-point response a different version of Fig. S8 (including a “Metabolites” group) is shown compared to the manuscript.,

We apologise for this. The manuscript version is the final version.

Suggestions:

a. Create a graphical abstract illustrating the suggested concept and interactions of key mechanistic players starting from SQ impact on MSC membrane stiffness, highlighting the mechanistic links to MSC immunomodulation factors.

This would not only help readers to follow better the authors' concept and to contextualize the data supporting their concept, but also show the authors where they could add evidence to support their claims and, thus, to the tap the (possible) full potential of their findings.

Please see new figure 7. We thank the reviewer for this useful suggestion.

Fig 7. Schematic illustrating main findings. (a) Culture on the SQ nanotopography causes a lowering of cell adhesion, cytoskeletal tension and mitochondrial association with actin microfilaments. This results in use of adenine, niacinamide, glutamic acid and citrate by the cells to drive glycolysis resulting in reduced T cell proliferation. (b) Addition of the glycolysis-driving metabolites into large scale MSC culture results in similar adhesion and intracellular tension reduction as seen when the cells are cultured on the SQ nanotopography along with secretion of TGF β , PGE-2 and IL-8 and resultant reduced T cell proliferation. Red cells = MSCs, green cells = T cells.

b. Clearly identify the molecular link(s) how changes in cell tension impact oxidative glycolysis and, if proven, immunomodulation.

We have performed new inhibition experiments that point to IDO-1 as a regulator.

c. Validate the MSC immunomodulation potential at least in one in vivo model as this is

currently still based solely on a single in vitro assay, i.e. suppression of PBMNC proliferation.

As noted above, we have discussed this with the journal.

Again, we thank the reviewer for their further consideration of our work and we very much hope that they now find our article acceptable for Nature Communications.

REVIEWERS' COMMENTS

Reviewer #3 (Remarks to the Author):

The authors have substantially increased evidence that, despite some prevailing inconsistencies that may lie also in the variability of MSC preparations, support their hypotheses and provide novel insights into the mechanisms of MSC immunomodulation and their potential applicability. Such, the reviewer's concerns and suggestions were addressed satisfactory.